# How to improve the state-of-the-art in metocean measurement datasets

Erik Quaeghebeur[*] and Michiel B. Zaaijer[*]

[*]Wind Energy Section, Delft University of Technology, Kluyverweg 1, 2629 HS Delft, the Netherlands

**Correspondence:** Michiel Zaaijer (M.B.Zaaijer@tudelft.nl)

**Abstract.** We present an analysis of three datasets of 10-minute metocean measurement statistics and our resulting recommendations to both producers and users of such datasets. Many of our recommendations are more generally of interest to all numerical measurement data producers. The datasets analyzed originate from offshore meteorological masts installed to support offshore wind farm planning and design: the Dutch OWEZ and MMIJ, and the German FINO1. Our analysis shows
that such datasets contain issues that users should look out for and whose prevalence can be reduced by producers. We also present expressions to derive uncertainty and bias values for the statistics from information typically available about sample uncertainty. We also observe that the format in which the data is disseminated is sub-optimal from the users' perspective and discuss how producers can create more immediately useful dataset files. Effectively, we advocate using an established binary format (HDF5 or netCDF4) instead of the typical text-based one (comma-separated values), as this allows for the inclusion of
relevant metadata and the creation of significantly smaller directly accessible dataset files. Next to informing producers of the advantages of these formats, we also provide concrete pointers to their effective use. Our conclusion is that datasets such as the ones we analyzed can be improved substantially in usefulness and convenience with limited effort.

**Key words:** metocean data, measurements, wind energy, dataset analysis, binary format, uncertainty, best practices

## 1 Introduction

The planning and design of offshore wind farms depends heavily on the availability of representative meteorological and ocean or 'metocean' measurement data. For example, the wind resource (the wind speed and direction distribution) at the candidate farm location is used to estimate energy production over the farm's lifetime and information about ocean waves is needed for wind turbine support structure design and planning installation & maintenance.

The data is collected by instruments placed on fixed offshore platforms, met masts, or measurement buoys deployed in
measurement campaigns. These campaigns are ordered by the project owner (a government or a farm developer) and set-up and carried out by contractors (applied research institutes or companies). The dataset producer (one or more of the contractors) collects and processes the data generated in these campaigns and provides it to dataset users. The datasets produced are often available publicly to these users, although usually with some access and usage restrictions, especially for commercial purposes.

We became interested in evaluating metocean measurement datasets after encountering a number of issues in a specific
dataset, both in data quality as well as in the dissemination format. (Our concrete purpose was to use it for wind farm en-

ergy production estimation.) Discussion with other users of such datasets showed that many found the typical dissemination approach, providing multiple files with comma-separated values, to be inconvenient or even a hindrance to their application. Most were not aware of the data quality issues we encountered, which can be categorized as faulty data, missing documentation, inappropriate statistic selection, limited data quality information, and suboptimal value encoding.

Therefore, we performed a study of three commonly used metocean datasets to answer essentially the following questions: (i) Are these issues commonly shared in metocean datasets? (ii) How can the issues that are present be addressed? This paper reports the results of that study. In brief: (i) Yes, there are shared issues, but, not unexpectedly, not all of them in all datasets. (ii) Dataset producers can address the issues with a few non-burdensome additions to their creation practice. Next to providing arguments for and detailing these conclusions, this paper is meant to raise awareness of the issues mentioned by giving concrete

examples. Furthermore, it provides dataset producers with concrete ideas about how to achieve substantial improvements with reasonable effort.

The users of the produced datasets are of course the farm developers, but also the academic world, whose usage is not necessarily restricted to wind energy applications. The context of our academic research is offshore wind energy, but the work we present here is relevant outside that area as well. Therefore, we treat all measured quantities on equal footing and do not

focus on wind and wave data. When our discussion goes beyond the analysis of the specific datasets we considered, it is also mostly independent of their metocean nature, but generally applies to any numerical time series data.

We structure the paper into two main sections. We start with an essentially descriptive Sect. 2, to give an overview of the datasets we considered and to identify the issues we encountered. The original contributions here are our thorough description, in-depth analysis, and expressions for the uncertainties and bias for the statistics' values that make up the datasets. In this

section we also mention options for addressing issues described, where it can be done compactly and where we believe it adds value for dataset producers. In the instructional Sect. 3 we discuss how the format of these datasets can be improved and thereby disseminated more conveniently. This section includes an up-to-date evaluation of binary dataset file format functionality. The recommendations to project owners, dataset producers, and users that follow from these analyses are collected at the end of this paper (Sect. 4), preceding the overall conclusions (Sect. 5).

## 2    The Datasets and Their Analysis

We split our discussion of the datasets into two parts: first, in Sect. 2.1, we present the three datasets in terms of context and content, then, in Sect. 2.2, we go over the issues we encountered.

### 2.1    A First Look at the Datasets

All three datasets we consider come from measuring masts in the North Sea and contain multiple multi-year 10-minute statistics

data, called 'series'. These 10-minute statistics are derived from higher-frequency measurements, called 'signals', of quantities measured by various instruments at various locations on the mast. The available statistics are the sample minimum, maximum, mean, and standard deviation.

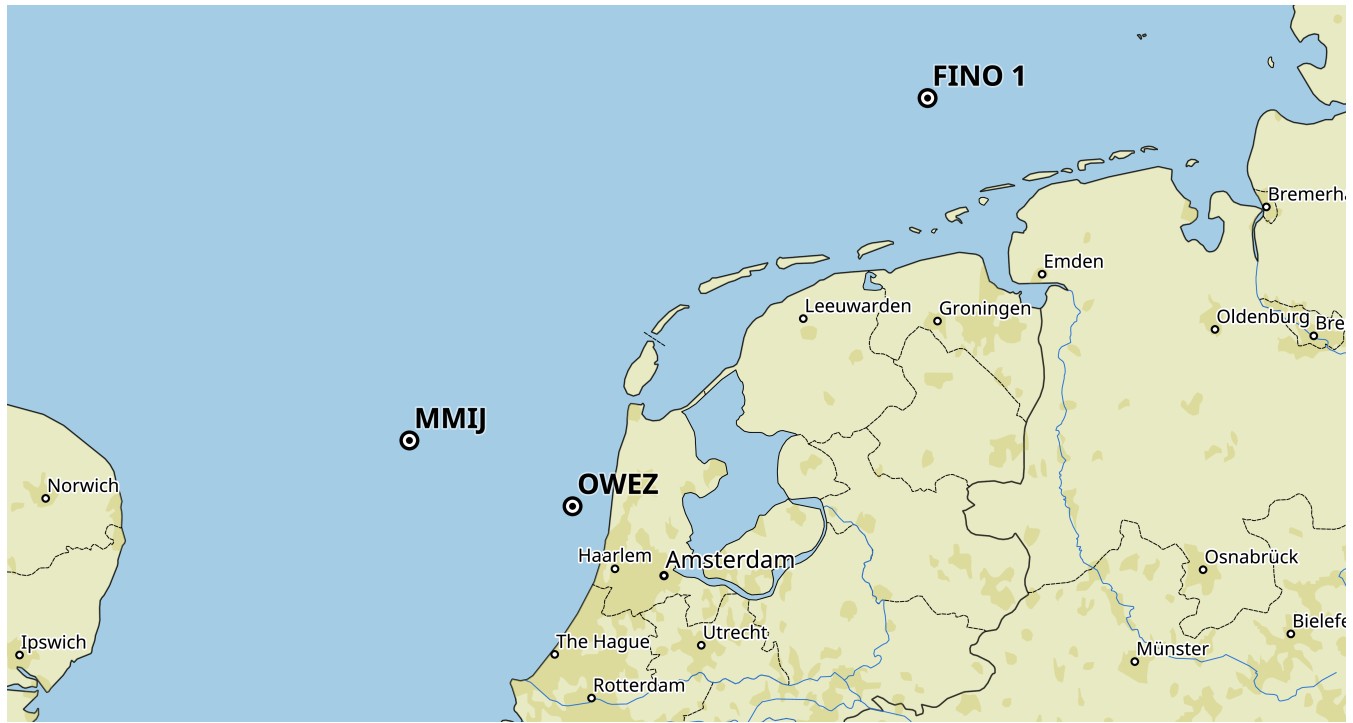

**Figure 1.** A map with the location of the three offshore met masts from which data was analyzed: OWEZ, MMIJ, and FINO1.

For each dataset, we give a brief description of the measurement site and setup, list the measurement period and quantities measured, describe the dissemination approach, point to available documentation, and highlight some further important aspects. We do this in full detail here for the first dataset, but for the other two we put aspects that are not substantively different in Appendix A1. We also provide a brief FAIRness analysis (Wilkinson et al., 2016) of the datasets in Appendix A1.3.

5     Common to all three datasets is that they can be downloaded from a website, where some documentation is available. But, also for all three, we needed to look up external sources and contact parties involved in the dataset creation process to get a more complete view. The collected metadata is available as part of a separate bundle (Quaeghebeur, 2020). It also includes details not mentioned in this paper, such as the make and type of instruments and loggers.

### 2.1.1    OWEZ — Offshore Windfarm Egmond aan Zee

10   To gather data before and after construction of the Offshore Windfarm Egmond aan Zee (OWEZ; *Offshore windpark Egmond aan Zee* in Dutch), a met mast was built on-site. Its location is $52°36'22.9''$ North, $4°23'22.7''$ East (WGS 84), which is 15 km off the Dutch coast near the town Egmond aan Zee. The location is indicated in Fig. 1. The mast was erected in 2003 and construction of the wind farm started in 2006. Data is publicly available for the period July 2005–December 2010. The instruments used and quantities measured, and some of their characteristics are listed in Table 1.

**Table 1.** An overview of the instruments and their locations on the OWEZ met mast (height in meters above mean sea level and boom orientation), the quantity measured, measurement uncertainty, the measurement ranges, and the sampling frequencies.

| Instrument (#) | Height[ah] [m] | Orientation[ao] | Quantity | Unit | Uncertainty[m] abs. | rel. [%] | Range[m] | Freq.[m] [Hz] |
|---|---|---|---|---|---|---|---|---|
| accelerometer (1) | 116 | mast | N-S accel. W-E accel. | $\mathrm{m\,s^{-2}}$ | 0.01 | | −30–30 | 33 |
| cup anemometer (9) | all | all | hor. wind sp. | $\mathrm{m\,s^{-1}}$ | 0.5 | | 0–50 | 4 |
| ultrasonic anemometer (3) | all | NE | hor. wind sp. vert. wind sp. | $\mathrm{m\,s^{-1}}$ | 0.01 | 1.5 | 0–60 | 4 |
| | | | wind direction | ° | 2 | | 0–359 | 4 |
| wind vane (9) | all | all | wind direction | ° | 1.4 | | 0–360 | |
| barometer (1) | 20 | mast | atm. pressure | mbar | 0.5 | | 600–1100 | |
| thermometer[i] (3) | all | S | ambient temp. | °C | 0.1 | | −40–80 | |
| hygrometer[i] (3) | all | S | rel. humidity | % | 1 | | 0–100 | |
| precipitation sensor (2) | 70 | NE, NW | precip. level | – | | | | |
| thermometer (1) | -3.8 | mast | water temp. | °C | 0.15 | 0.1 | -180–600 | |
| acoustic wave and current profiler[f] (1) | -17 | ? | water temp. | °C | 0.1 | | −4–40 | 1 |
| | | | water level | m | | | | 4 |
| | | | wave height | m | 0.01 | 1 | −15–15 | 4 |
| | | | wave direction | ° | 2 | | 0–359 | 2 |
| | | | wave period | s | | | 0.5–50 | 2 |
| | | | current vel. 7 m current vel. 11 m | $\mathrm{m\,s^{-1}}$ | 0.005 | 1 | -10–10 | 1 |
| | | | current dir. 7 m current dir. 11 m | ° | | | 0–359 | 1 |

[ah] For height, 'all' corresponds to 21 m, 70 m, 116 m.

[ao] For orientation, 'all' corresponds to NE, NW, S or -60°, 60°, and 180°, respectively (North corresponding to 0°).

[i] Thermometer and hygrometer are contained in a single package.     [f] The given sampling frequencies are upper bounds.     [m] Missing values are unknown.

Due to an agreement between the Dutch government and the OWEZ developer, data gathered and reports written in the context of the wind farm's construction have been made publicly available. This is done through a website where these materials can be downloaded (NoordzeeWind, 2019). The metocean dataset can be downloaded as 66 separate monthly, compressed Excel (xls) spreadsheet files. The total size is almost 1 GB, or about 400 MB compressed. This represents data points for 289 296 10-minute intervals. The data in each file is structured as follows:

– 6 date-time columns (year, month, day, hour, minutes, seconds);

– 48 'channels' of five columns each: an integer identifier 'Channel' and four real-valued statistics, 'Max', 'Min', 'Mean', and 'StdDev'; each channel corresponding to a specific measured quantity and location on the mast.

In the Excel files, the statistics' values are encoded as 8-byte binary floating point numbers.

Information about the dataset, the met mast, and its context is available through the same website. In particular, there is a user manual (Kouwenhoven, 2007) and several reports from which further information can be learned (e.g., Curvers, 2007; Eecen and Branlard, 2008; Wagenaar and Eecen, 2010a, b). Information about the instruments used and in particular the measurement uncertainty had to be looked up in specification sheets or obtained through personal communication with people involved in the project (cf. Acknowledgements).

### 2.1.2 MMIJ — Measuring Mast IJmuiden

The second dataset, 'MMIJ', comes from a met mast in the Dutch part of the North Sea. The location is indicated in Fig. 1. Details can be found in Appendix A1.1.

The exact set of signals differs of course from the OWEZ dataset; we have given an overview in Table A1 in the appendix. The data was collected during the period 2011–2016, a period of time comparable in length to OWEZ. The dataset is made available as a single semicolon-separated values (csv) file and the statistics' values are encoded in a decimal fixed-point format with five fractional digits (x...x.xxxxx).

### 2.1.3 FINO1 — Research Platform in the North Sea and the Baltic Sea Nr. 1

The third dataset, 'FINO1', comes from a met mast in the German part of the North Sea. The location is indicated in Fig. 1. Details can be found in Appendix A1.2.

The exact set of signals again differs from the OWEZ dataset; we have given an overview in Table A2 in the appendix. The data investigated was collected during the period 2004–2016, so a period of time more than twice as long as for the other two datasets. A difference with the other two datasets is that not all statistics are available for all signals. Also, it is free for academic research purposes, but not for commercial use, in contrast to the two other datasets. The dataset is made available as a set of tab-separated values (dat) files and the statistics' values are encoded in a decimal fixed-point format with up to two fractional digits (x...x.xx). For each quantity, a quality column is included next to the statistics' columns.

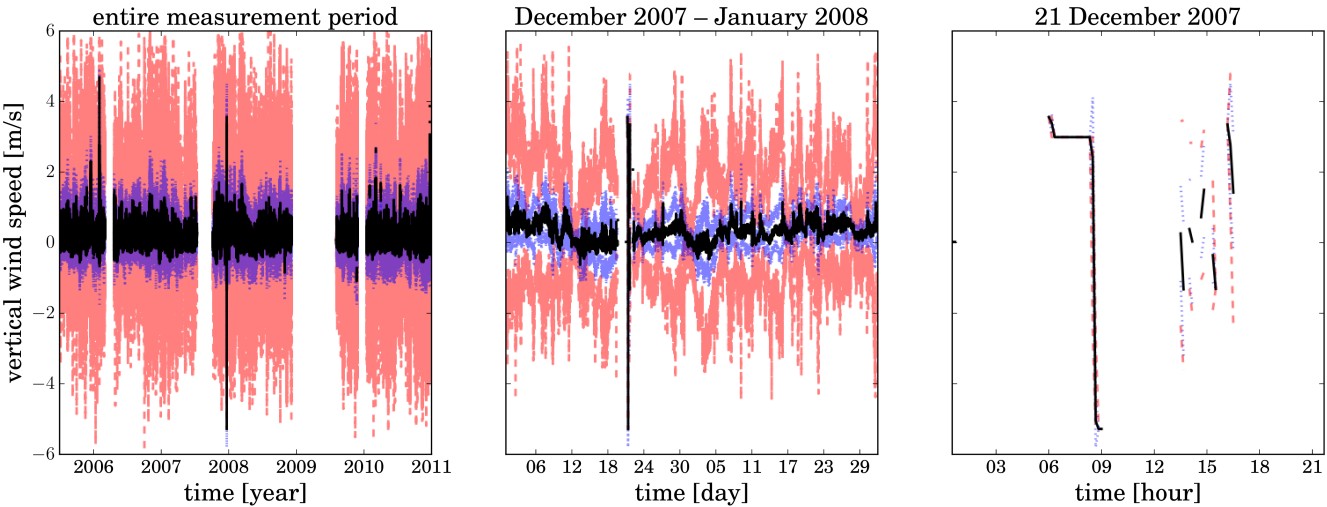

**Figure 2.** An illustration of the visual inspection and zooming of plots. We present the OWEZ vertical wind speed data collected by the ultrasonic anemometer at the NE-116 m location. (Mean in black; mean $\pm$ standard deviation in blue; minimum and maximum in red.)

## 2.2 Dataset Issues

We split the issues encountered in the datasets into five categories each discussed in their own section: faulty data (Sect. 2.2.1), documentation (Sect. 2.2.2), statistic selection (Sect. 2.2.3), quality flags (Sect. 2.2.4), and value encoding & uncertainty propagation (Sect. 2.2.5).

### 2.2.1 Faulty Data

It is not unusual that the measured signals (raw data) contain faulty data. With this we mean data values that cannot correspond to the actual values, or are very unlikely to correspond to them. The dataset producers deal with such faulty data, e.g., by flagging or removing it, when creating the datasets of statistics series we study. Nevertheless, each of the three datasets presented above contained remaining faulty data. We stumbled upon initial examples, but then systematically looked for issues.

To facilitate this systematic and partly automated investigation, we created binary file format versions of the datasets (HDF5 format for OWEZ and netCDF4 format for MMIJ and FINO1) in which metadata such as range and possible values can be stored alongside the data itself. We discuss these formats in more detail in Sect. 3. The automation essentially consisted of looping over all signals and statistics to detect issues; further investigation was done manually.

Concretely:

1. We performed interactive visual inspection of plots of the individual datasets, including zooming in on suspicious-looking parts. Figure 2 provides an example. The plots should be read as follows: the mean value is given by the 'inner'

**Table 2.** An overview of the (largest) range violations present in the FINO1 dataset. (Values rounded to three digits.)

| Instrument | Quantity | Unit | Statistic | Lowest | Instr. range | Highest |
|---|---|---|---|---|---|---|
| cup anemometer | hor. wind. sp. | $\mathrm{m\,s^{-1}}$ | min | 0.0313 | 0.1–75 | |
| | | | max | | 0.1–75 | 1690 |
| ultrasonic anemometer | hor. wind. sp. | $\mathrm{m\,s^{-1}}$ | max | | 0–45 | 45.6 |
| | wind direction | ° | avg | | 0–359 | 360 |
| wind vane | wind direction | ° | max | | 0–360 | 521 |
| | | | avg | | 0–360 | 366 |
| barometer | atm. pressure | hPa | avg | 0.003 91 | 800–1060 | |
| hygrometer | rel. humidity | % | avg | 0.0313 | 10–100 | 102 |
| precipitation sensor | precip. intensity | mA | avg | 0.001 95 | 4–20 | 45.3 |
| pyranometer | global radiation | $\mathrm{W\,m^{-2}}$ | avg | −4.86 | 0–4000 | 145 000 |

full (black) line; mean values plus and minus one standard deviation are given by the 'intermediate' dotted (blue) lines; minima and maxima are given by the 'outer' dashed (red) lines.

The plots in this figure are snapshots of an interactive visualization procedure: Even though the lines overlap in the unzoomed left-hand plot, an anomalous extreme mean value is visible around the 2007–2008 year change. Zooming in a bit gives the middle plot, where the statistics start becoming visually separated and where the anomaly stands out even more. Zooming in further gives the right-hand plot, which shows that many missing values surround the anomaly, further suggesting that the values still present here may not be reliable. (We do not know *why* the surrounding values are missing.)

2. We ran automated checks for values outside the instrument's range for the series or for inconsistent sets of statistics' values. Let us clarify what inconsistent sets of statistics' values are. Statistic values imply bounds on the value of other statistics. If such a constraint is violated for some 10-minute interval, the tuple of statistics (minimum $\check{x}$, maximum $\hat{x}$, mean $\bar{x}$, standard deviation $s_x$) for that interval is inconsistent. For example, it should be the case that $\check{x} \leq \bar{x} \leq \hat{x}$; violations of this constraint are present, e.g., in the FINO1 cup anemometer wind speed data. Less obvious constraints involving the sample standard deviation also exist. We used $\frac{1}{2}|\hat{x}-\check{x}|$ as the general upper bound for the standard deviation, given that the values lie in the interval $[\check{x}, \hat{x}]$ (Shiffler and Harsha, 1980). (Here $\check{x}$ and $\hat{x}$ can be replaced by range bounds in case the minimum and maximum statistics are not present in the dataset.) Any such inconsistency is a serious issue, as it indicates a deficiency somewhere in the procedures for calculating statistics and their post-processing.

As an example, the range violations in the FINO1 dataset gave the results listed in Table 2. Some range violations point to faulty data (e.g., cup anemometer-hor. wind speed-max, where the value exceeds the bound by more than an order of magnitude), others suggest a need for more elaborate uncertainty analysis (e.g., hygrometer-rel. humidity-avg, where

the violating values probably correspond to the bounds) or more elaborate handling of the range bounds (e.g., wind vane-wind direction-max, where the upper *bound* could be increased; cf. also Appendix A2.1).

The code producing the results of Table 2 is publicly available (Quaeghebeur, 2020). The fact that our netCDF4 version of the dataset is (uniformly) structured and contains metadata allows the code to be generic, i.e., not variable-specific, and therefore compact.

3. We did checks of the occurring values, for quantities with a discrete number of possible values. One example are the synoptic code 'Max' values from the MMIJ precipitation monitor. The check showed the following values to be present:

| $-998$ | $-997$ | $-953$ | $-952$ | $-950$ | $-900$ | $-176$ | $-16$ | | | | | | | | | | | |
|---|---|---|---|---|---|---|---|---|---|---|---|---|---|---|---|---|---|---|
| 0 | 51 | 53 | 55 | 58 | 59 | 61 | 63 | 65 | 68 | 69 | 71 | 73 | 75 | 77 | 87 | 88 | 89 | 90 |
| 108 | | | | | | | | | | | | | | | | | | |

Synoptic code values below 0 and above 99 do not exist (World Meteorological Organization, 2016, p. 356–358), so faulty data is present here. Only integer values are present here, but erroneous fractional values would also be detected.

The code for performing this check is publicly available (Quaeghebeur, 2020).

4. We ran automated checks for outlier candidates. There can be both 'classical' outliers, i.e., values outside the range typical for that series, and 'dynamic' ones, i.e., subsequent value pairs whose difference ('rate-of-change') lies outside the difference typical for that series's time-variation. Both types of outliers can, but do not necessarily correspond to faulty data.

In further manual analysis of outlier candidates, causes may be identified, providing feedback on the data collection and processing procedures. For example, both in the MMIJ and FINO1 datasets, we encountered sudden drops to the value zero for some series *at regular time instances*; this quite likely corresponds to foreseeable or detectable sensor resets of some kind.

There are many methods for outlier detection (Aggarwal, 2017). But, in this paper, we just wish to point out that there is a clear need for some form of outlier detection to be used in the creation of metocean 10-minute statistics datasets. Namely, the datasets we analyzed would benefit enormously from even a basic analysis; we suspect this generalizes to other such datasets produced in the wind energy field. To make this need apparent, we present a set of plots in Figs. 3–6 that illustrate that indeed there are still outliers present in the datasets. We devised this type of plot as an alternative to lag-1 plots (which plot $x_{k+1}$ versus $x_k$), so that rate-of-change magnitudes can be read off directly.

These plots, of which examples are given in Figs. 3–6, should be read as follows: The horizontal '$x$'-axis shows measurement value; the vertical '$y$'-axis shows the absolute value of the mean of the differences with the preceding and next measurement values. Each dot corresponds to a measurement. Lines connect successive measurements. Only those measurements are shown with an $x$-percentile outside $[0.1, 99.9]$ or a $y$-percentile above 99, so the brunt of the measurements are not shown. (These bounds are somewhat arbitrary, but reasonable for the size of the datasets.) The $y$-axis is linear

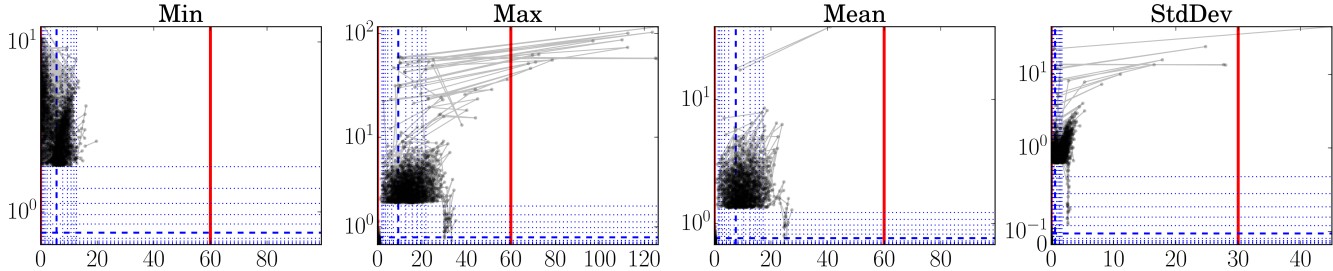

**Figure 3.** Illustrative plots for visually identifying outliers (cf. pages 8–9 for an explanation): OWEZ 21 m NW ultrasonic anemometer horizontal wind speed data $[\mathrm{m\,s^{-1}}]$.

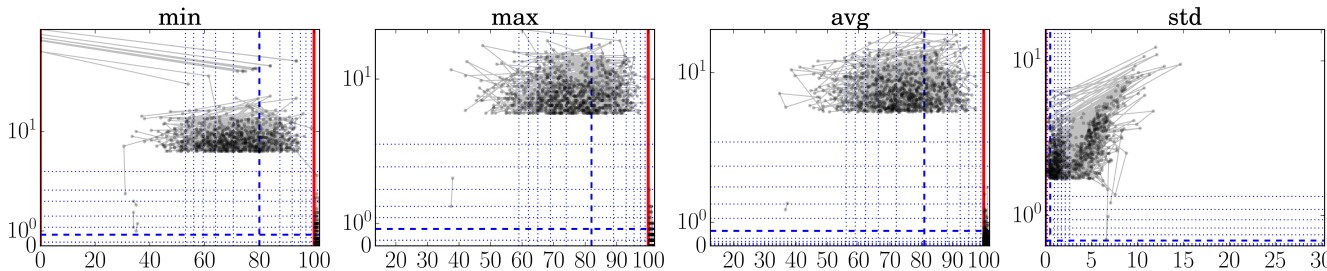

**Figure 4.** Illustrative plots for visually identifying outliers (cf. pages 8–9 for an explanation): MMIJ 21 m relative humidity data [%].

until the 99th percentile, and logarithmic above. To give an idea about the distribution of all the measurement points, so also the ones that are not shown, we add (blue) lines for specific fractiles: thick dashed for the median and thin dotted for $\{\frac{1}{2^6}, \ldots, \frac{1}{8}, \frac{1}{4}, \frac{3}{4}, \frac{7}{8}, \ldots, 1 - \frac{1}{2^6}\}$. Thick full (red) lines are added as necessary to indicate range bounds.

In Fig. 3, there are some suspiciously high values, some even beyond the nominal measurement range of the instrument.

5 This is also the case for the 'Min' and 'Mean' statistics, even if the probably isolated responsible data points are not visible. In Fig. 4, there are suspicious 0% values and several of values beyond 100%. In Fig. 5, we see a cluster of data points at suspiciously low values and some impossibly fast 10-minute pressure changes, a number of them more than 100 hPa. In Fig. 6, we see a quite large number of atypically high temperatures and some impossibly fast 10-minute temperature changes, a couple of them of more than 30 °C.

10 Outlier plots for all data series are available as supplementary material for this paper. The code producing them is publicly available (Quaeghebeur, 2020).

Our analysis was generic in the sense that we did not make use of quantity-specific domain knowledge (e.g., empirical relationships between mean and maximum) or measurement setup-specific knowledge (e.g., met mast influence on wind speed). In the context of wind resource assessment, Brower (2012) gives a description of a data validation procedure that does take into

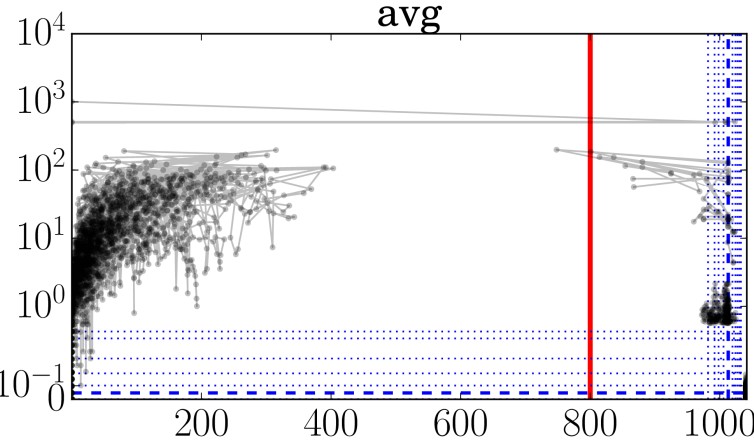

**Figure 5.** Illustrative plots for visually identifying outliers (cf. pages 8–9 for an explanation): FINO1 21 m air pressure data [hPa].

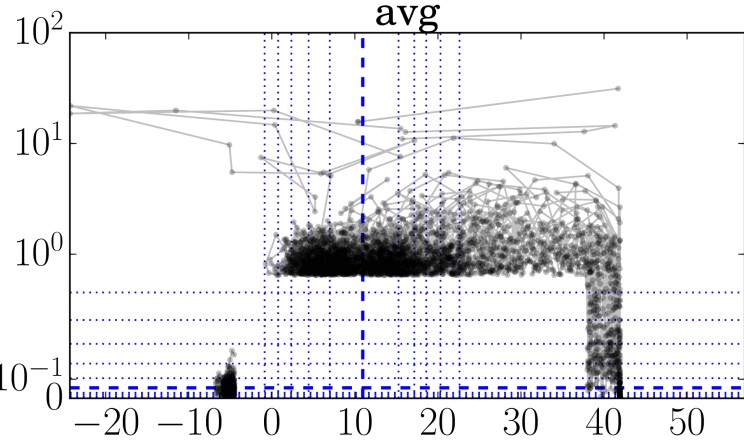

**Figure 6.** Illustrative plots for visually identifying outliers (cf. pages 8–9 for an explanation): FINO1 72 m ambient temperature data [°C].

account such specifics. Meek and Hatfield (1994) proposed signal-specific rules for checking meteorological measurements for range violations, rate-of-change outliers, and no-observed-change occurrences.

For all of the issues presented in this section, the dataset producer is better placed to interpret them, given that they have information about the data acquisition and processing procedures that the user lacks. Therefore it is the dataset producer who would ideally identify such issues and fix them, if possible, or otherwise at least mask or flag them. Given, as illustrated, the relative simplicity of the required analyses, relatively little effort may be required for a substantial increase in dataset quality.

### 2.2.2 Documentation

As mentioned in Sect. 2.1, for each of the three datasets we investigated, documentation on the measurement setup, instruments, and quantities measured is available. Usually, this takes the form of a website, data manual, overview table, or a combination thereof. However, for purposes of interpretation and use of these datasets, some essential or potentially useful information is often missing.

We consider the information we listed in the overview Tables 1, A1, and A2 to be essential: instrument location, quantity measured, its unit, information about accuracy (e.g., by giving absolute and relative uncertainty),[1] range, and, given our focus on statistics data, sampling frequency. For categorical data such as binary yes/no sensors (e.g., precipitation presence) or enumeration values (e.g., synoptic codes), range is of course replaced by a set of possible values and unit by a description of how to interpret those possible values.

How do the three datasets fare in terms of documentation?

**Timestamps** All data values are accompanied by timestamps spaced ten minutes apart. However, for none of the three datasets it is mentioned whether this time-stamp refers to the time of the first, last, or even some other sample. Knowing this is necessary for the precise combination of datasets. If we assume that the samples underlying the dataset start at the full hour, which corresponds to the raw data we have seen for OWEZ, we can deduce the convention used. Based on whether the first time-stamp in a data file has '00' or '10' for its minutes value, we assume that OWEZ and MMIJ are first-sample based and FINO1 is last-sample based.

**Location** For all three datasets, the documentation about location was good to excellent: technical drawings of the mast with instrument locations or detailed data about orientation and height. (Pictures or video footage would of course further increase confidence in the accuracy of the drawings.) A small comment we can make here is that the location information in the series names used sometimes does not directly correspond to the actual situation. For example, in the MMIJ dataset a $46.5°$ angle offset of boom orientation relative to the (geographic) North needs to be accounted for and in the FINO1 dataset some height labels differed from the documented heights.

---

[1]We follow the Joint Committee for Guides in Metrology (2012) in our usage of '(measurement) accuracy' and '(measurement) uncertainty'. Namely, the former refers to a qualitative description of the "closeness of agreement between a measured quantity value and a true quantity value of a measurand" and the latter to a quantitative measure, i.e., a "non-negative parameter characterizing the dispersion of the quantity values being attributed to a measurand, based on the information used". These terms cover both systematic and random aspects.

**Quantities & units**  The description of the actual quantities measured and their units was in general also quite good. There were two clear exceptions: (i) The precipitation detector was completely omitted from the MMIJ documentation. (ii) Precipitation data from FINO1 at 23 m contained the concatenation of both presence (yes/no) and intensity data. Also, the interpretation of binary codes (e.g., does 0 correspond to yes or no?) was for none of the datasets explicitly given, but had to be deduced from the data.

**Ranges**  Ranges and sets of possible values were mostly left unmentioned in the documentation, except for those available in instrument data-sheets included in the OWEZ and MMIJ data manuals. Making the data sheets of the instruments available in such a way turned out to be convenient, as tracking them down is in our experience not always possible.

**Accuracy**[1]  Accuracy information was available in the FINO1 overview table and for those instruments for which the data sheet was included in the OWEZ and MMIJ data manuals. For the other signals, we had to rely on the information found in data sheets not available in the datasets' documentation or website. Entirely absent is a discussion of the impact on accuracy of all other aspects of the measurement setup (e.g., analog-to-digital conversion) and data processing (e.g., the application of calibration factors). Such a discussion would allow researchers using the datasets to get a more complete picture of the accuracy of the values in the datasets.

**Sampling frequency**  The sampling frequencies were available in the documentation for MMIJ and FINO1, but not for OWEZ. This information is essential for the estimation of the uncertainty of the mean and standard deviation statistics (cf. Sect. 2.2.5).

**Instruments & their settings**  We mentioned our use of data sheets a few times before. To find these when they are not included in the documentation, the exact instrument models need to be available. This was the case for all three datasets. However, this may not be enough: the measurement characteristics of some instruments (e.g., barometers) depends on specific settings, especially when they perform digital processing. These settings were never described. Furthermore, loggers are an essential piece of the measurement chain and therefore need to be documented as well. For MMIJ and FINO1 this is the case, but not for OWEZ.

**Data processing**  Next to its relevance for assessing the accuracy of the values in the dataset, a good view of the data processing pipeline is important for other aspects as well:

- When is data considered to be faulty and flagged in or omitted from the dataset accordingly? This is entirely missing for OWEZ and FINO1, but some information is given for MMIJ: if some values in a 10-minute interval are missing, the corresponding statistics are marked as missing. How faulty data values are encoded is documented for OWEZ (as the value $-999\,999$), but not for MMIJ and FINO1. For MMIJ, the convention used (the string 'NaN') seems to be used quite consistently, although some precipitation monitor outlier values might actually be other markers for faulty data. For FINO1, there are two main faulty data placeholder values easily identified from the datasets: $-999.99$ and $-999$. However, other values are also present, such as 0 and variants of the two main ones, such as $999$, $-999.9$ and $-1000$.

– How are the statistics calculated? This is never mentioned in the documentation. For most signals not much ambiguity can arise, as there is not much choice, being limited to a possible bias correction approach for the standard deviation. However, for directional data, it is very much pertinent which definition of mean and standard deviation have been used: arithmetic or directional mean, classical or circular standard deviation (see, e.g., Fisher, 1995).

– Do the data processing steps to arrive at the statistics have any weaknesses, numerical or other? For example, in the FINO1 wind speed data, there appear max values that, suspiciously, are a factor ten or hundred times larger than the surrounding values. Leaving such things unexplained severely reduces the trust in the dataset.

It is clear from the above list that while already a good amount of information is available, quite a number of very useful pieces of information are missing. Many of these are available to the dataset producers, so again the quality of the datasets, now in terms of documentation, can be substantially improved with little effort relative to the whole of the measurement campaign.

Unmentioned as of yet is that essentially all the documentation for these datasets is provided in a way accessible to humans, but not in a machine-readable way. Much of the information described in the documentation can however be encoded as metadata in a standardized and machine-readable way. Metadata is discussed further in Sect. 3.1.

### 2.2.3 Statistic Selection

As seen in the overview sections 2.1.1, 2.1.2, and 2.1.3, for all three datasets the statistics provided are essentially the same: minimum, maximum, mean, and standard deviation. Only for FINO1 not all statistics are included for all quantities. In this section, we are going to discuss these statistic selection choices, pointing out issues that arise from them.

The uniformity of the statistics provided is convenient when reading out the data, as it reduces the user's quantity-specific code. However, when the signal's values do not represent an (underlying) linear scale providing the minimum, maximum, mean, and standard deviation does not make much sense; it may actually cause misinterpretation. This is usually the case for categorical signals, such as the MMIJ synoptic code signal. In such cases, other statistics must be chosen. For example, for binary quantities such as yes/no precipitation data, giving the relative frequency of just one of the two values captures all the information present in the typical set of four statistics.

As said, in the FINO1 dataset statistics are sometimes omitted, but mostly for other reasons. For quantities that are considered to be 'slow-varying' (such as atmospheric pressure, ambient temperature, and relative humidity) only the mean has been recorded.[2] However, next to the convenience of uniform sets of statistics, having multiple statistics for a measurement interval is useful for data quality assessment. (Possible storage and transfer constraints are of course valid reasons for limiting the number of statistics.) For directional quantities such as wind direction the minimum and maximum were omitted because these are considered meaningless by the dataset producer.[2] The OWEZ and MMIJ datasets show, however, that it is possible to give meaningful definitions of maximum and minimum for directional data. (See Appendix A2.1 for a concrete approach.) This can be valuable information, as it makes it possible to deduce, e.g., the sector extent from which the wind has blown during a time interval.

---

[2]Personal communication d.d. 2017-06-27 with Richard Fruehmann (cf. Acknowledgements).

### 2.2.4 Quality flags

Next to statistics, we saw in Sect. 2.1.3 that the FINO1 dataset also contains a categorical quality flag for each set of statistics. Such information is not present in the other two datasets.

Including such a flag makes it possible to also provide information about missingness, i.e., to indicate why one or more statistic values is missing at that time instant. Such information is often encoded using a bit field, i.e., a binary mapping from quality issues and missingness mechanisms to true (1) and false (0); this bit field can be recorded as a positive integer. For example, consider the following tuple of quality issues and missingness mechanisms: ('suspect value jumps', 'out-of-range values', 'unknown missingness mechanism', 'icing', 'instrument off-line'). Then the bit string '00000' (or integer 0) would denote a measurement interval without any (identified) issues and for example '010010' (or integer 18) would correspond to a measurement interval with both instrument icing and out-of-range values detected.

Of course other information next to missingness mechanisms can be included in the quality flag bit field, also for non-missing values, as is done for FINO1. For example, this can be used to indicate possibly faulty data (cf. Sect. 2.2.1) that has not been removed (made missing).

### 2.2.5 Value Encoding & uncertainty propagation

In the overview sections 2.1.1, 2.1.2, and 2.1.3, for all three datasets, the values themselves are encoded as fixed-point values for MMIJ and FINO1 and as a binary floating point double for OWEZ. There is, however, more to be said about what exactly is encoded and which information can be reflected in the encoding. We do that here.

Signal values have a natural set they belong to. Relative humidity, for example, is a fraction, i.e., a value between zero and one. Categorical signals take values in a predefined enumerated set. If for such signals values are given outside of this set, this is a source of confusion: the user may wonder whether they can just round erroneous values to the nearest enumerated one or treat them as faulty. For example, the MMIJ precipitation detector's precipitation presence signal contains values *around* the enumerated ones and its precipitation monitors' precipitation presence signals contains values far outside the range of enumerated values. Another case are continuous signals that are at one point expressed as current or voltage values: the end user will be less certain about the correct translation procedure to the correct units than the data processor. For example, the FINO1 precipitation intensity signal is expressed as a current instead of an accumulation speed.

In the OWEZ and FINO1 datasets it sometimes occurs that certain statistics are marked as faulty or missing, while nevertheless other statistics for the same signal at the same instance are available. From inspection of such data, it is clear that it can both happen that the values of these other statistics seem reasonable or faulty. An explanation of why the data values are partly missing would preserve trust in the non-missing values. This requires a description of the processes creating such a situation (cf. Sect. 2.2.2), but could also include instance-specific information in a flag value (cf. Sect. 2.2.4).

The values stored in the dataset do not in general encode their accuracy. For the MMIJ and FINO1 datasets, values used a fixed-point format, but the number of decimal digits used is not directly related to the accuracy information available for the different quantities. This fact may be overlooked by users, resulting in possible misinterpretations.

To avoid misinterpretation, it is possible to add an estimate for a value's uncertainty, e.g., by rounding and specifying a corresponding number of significant digits. Accuracy information was only available for signal values (i.e., high frequency samples), typically as absolute uncertainties $\varepsilon_a$ and relative uncertainties $\varepsilon_r$. Below, we give expressions for propagating this information to the statistics, as this does not seem available in the literature, and discuss further factors affecting the statistics'

uncertainty. The nontrivial derivations of these expressions and a description of the underlying model for the measurement process can be found in Appendix A2.2. The most important assumption made in these derivations is that $\varepsilon_r^2 \ll 1 \ll n$, where $n$ is the number of samples per averaging interval.

Sample uncertainties can be propagated to the statistics of the $n$ signal values $x_k$ per averaging interval, which is 10 minutes for the datasets discussed in this paper. For this, we essentially assume independence and normality of the corresponding

uncertainties $\varepsilon_{x_k}$. Also the uncertainty in the statistics due to the finite nature of the samples can be quantified based on the fact that the sum appearing in the calculation of the mean and standard deviation can be seen as a simple form of quadrature. Let $\check{x}$ and $\hat{x}$ be the minimum and maximum values in the sample; let $\bar{x} = \frac{1}{n}\sum_{k=1}^{n} x_k$ and $s_x^2 = \frac{1}{n}\sum_{k=1}^{n}(x_k - \bar{x})^2$ be the sample mean and sample variance. We find the following expressions for the squared uncertainties of the statistics:

$$\varepsilon_x^2 \approx \left(\varepsilon_a^2 + \varepsilon_r^2 x^2\right) + \frac{1}{n^2}\delta^2 \ \text{ for } x \in \{\check{x},\hat{x}\}, \quad \varepsilon_{\bar{x}}^2 \approx \frac{1}{n}\left(\varepsilon_a^2 + \varepsilon_r^2(\bar{x}^2 + s_x^2)\right) + \frac{1}{n^2}\delta^2, \quad \varepsilon_{s_x}^2 \geq \frac{1}{n}\left(\varepsilon_a^2 + \varepsilon_r^2(\bar{x}^2 + 3s_x^2)\right) + \frac{1}{n^2}\delta^2.$$

Here $\delta \approx \frac{\hat{x}-\check{x}}{2}$; in case $\hat{x}$ and $\check{x}$ are unavailable, $\delta \approx z_{1-1/n}s_x$ can be used instead, where $z_{1-1/n}$ is the standard normal quantile for exceedance probability $1/n$. The uncertainty due to the finite sample size, the term $\frac{1}{n^2}\delta^2$, diminishes much faster as a function of $n$ than the uncertainty due to the measurement noise, expressed by the other terms. In practice, this second term is therefore negligible unless $\varepsilon_a$ and $\varepsilon_r$ are taken to be zero because no information is available about them.

Next to having associated uncertainties, the sample statistics can also be biased estimators of the statistics for the underlying

signal. It turns out that only the sample standard deviation $s_x$ is biased and that

$$s_x' = \sqrt{\max\{s_x^2 - \left(\varepsilon_a^2 + \varepsilon_r^2\bar{x}^2\right), 0\}}$$

would be a better estimate from this perspective.

To get a more concrete view of these uncertainties and bias, we provide average relative uncertainty and bias values for the MMIJ dataset in Table 3. (The code producing the results of this table is publicly available (Quaeghebeur, 2020).) The variation

of the uncertainties and bias is substantial, so this table of averages does not provide a complete picture, but enough to draw some conclusions:

– A fixed-point format does not have the flexibility to give the appropriate number of significant digits; usually either too many or too few are given.

– While the uncertainty is usually rather small (up to a few percent), in some cases it is substantial (around ten percent or

more).

– The bias in the sample standard deviation can in general not be ignored. (For example, for ambient temperature, we see that the bias-corrected value is smaller than the uncertainty.)

**Table 3.** Average relative uncertainties and bias in percent for quantities from the MMIJ dataset for which some (likely incomplete) uncertainty information is available. (See Table A1 for more information about the quantities. The values are given with two digits, but it is not implied that both are significant.)

| Instrument | Quantity | $\dfrac{\varepsilon_{\tilde{x}}}{\tilde{x}}$ | $\dfrac{\varepsilon_{\hat{x}}}{\hat{x}}$ | $\dfrac{\varepsilon_{\bar{x}}}{\bar{x}}$ | $\dfrac{\varepsilon_{s_x}}{s_x}$ | $\dfrac{s'_x}{s_x}$ | $1-\dfrac{s'_x}{s_x}$ |
|---|---|---|---|---|---|---|---|
| cup anemometer | hor. wind sp. | 4.3 | 2.7 | 0.067 | 1.0 | 81 | 19 |
| ultrasonic anemometer | wind sp. X dir. | 10 | 11 | 2.4 | 3.0 | 88 | 12 |
|  | wind sp. Y dir. | 11 | 12 | 2.4 | 3.0 | 89 | 11 |
|  | wind sp. Z dir. | 16 | 29 | 7.7 | 3.3 | 81 | 19 |
| wind vane | wind direction | 2.2 | 0.77 | 0.056 | 1.4 | 94 | 6.4 |
| barometer | atm. pressure | 0.017 | 0.0099 | 0.00021 | 7.2 | 5.2 | 95 |
| thermometer | ambient temp. | 2.3 | 2.3 | 0.067 | 35 | 4.4 | 96 |
| hygrometer | rel. humidity | 1.3 | 1.3 | 0.026 | 8.7 | 17 | 83 |
| precipitation monitor | precip. intensity | 17 | 17 | 0.45 | 3.4 | 98 | 2.0 |
| from[v,s] cup anemometer | hor. wind sp. | 2.7 | 1.7 | 0.044 | 0.76 | 89 | 11 |
| from[v] ultrasonic anemometer | wind sp. magn. | 3.9 | 3.0 | 3.0 | 3.4 | 97 | 3.5 |
|  | hor. wind sp. | 2.0 | 0.79 | 0.042 | 0.44 | 98 | 2.3 |
| from[v,s] ultrasonic anemometer | hor. wind sp. | 1.2 | 0.58 | 0.045 | 0.31 | 99 | 1.2 |
| from[v,s] wind vane | wind direction | 1.8 | 0.56 | 0.042 | 0.51 | 96 | 3.9 |
| from[v] barometer and thermometer | air density | 0.016 | 0.0085 | 0.00018 | 2.0 | 72 | 28 |

[s] Correction for tower shadow by selective averaging of values at the same height.

[v] Virtual measurement; namely, derived from signals obtained with one or more actual instruments.

What the impact of uncertainty and bias are depends on the application. (For example, turbulence intensity estimation is clearly affected by the bias in the wind speed sample standard deviation. Concretely $\mathrm{TI}'/\mathrm{TI} = \frac{s'_x}{\bar{x}} / \frac{s_x}{\bar{x}} = s'_x/s_x$ for horizontal wind speed; e.g., an average reduction of turbulence intensity up to about 20%.) But to be able to assess this impact, uncertainty and bias values must be available, making expressions such as the above essential.

Before closing this Section, it is important to stress that the expressions for propagated uncertainties and biases above are generic. Namely, their derivation does not depend on the specific quantity considered or instrument used. Detailed knowledge of the measuring instrument's properties may allow for better uncertainty estimates or additional uncertainty and bias terms. For example, for cup anemometers, it is known that there is a positive bias of 0.5%–8% in the mean wind speed, but that this bias can be greatly reduced using wind direction variance estimates (Kristensen, 1999). Also, the IEC 61400-12-1 standard prescribes how the wind speed uncertainty should be calculated for calibrated cup anemometers (IEC, 2017, App. F), which may lead to high-quality estimates for $\varepsilon_{\mathrm{a}}$ and $\varepsilon_{\mathrm{r}}$.

## 3 Dataset Formatting

We split our discussion of dataset file formats into two parts. First, in Sect. 3.1, we give an overview of the formats that are currently used for the dissemination of the datasets studied and existing alternatives that we argue to be superior. Then, in Sect. 3.2, we take a closer look at the potential of these alternatives based on our practical experience with them.

### 3.1 A Comparison of Dataset File Formats

We saw in Sect. 2.1, during our first look at the datasets we studied, that these were disseminated as a compressed set of Excel files for OWEZ, a compressed semicolon-separated values file for MMIJ, and a compressed set of tab-separated values files for FINO1. In the Excel files, the values are stored as 8-byte binary floating point numbers. In the delimiter-separated values files the values are specified in a fixed-point decimal text format, with five (MMIJ) and two (FINO1) fractional digits. All of these are essentially table-based formats, where columns correspond to series and rows correspond to values for a specific time instance. (This structure satisfies the requirements of 'tidy data' according to Wickham (2014), apart from being split over multiple files.) Some metadata is included in two or more header lines, such as series identifiers and the unit.

We created binary file format versions of the datasets; in HDF5 format (The HDF Group, 2019a) for OWEZ and in netCDF4 format (Unidata, 2018) for MMIJ and FINO1. Both formats are platform-independent. Files in netCDF4 format are actually HDF5 files, but adhering to the netCDF data model (Rew et al., 2006). The use of a different data model is reflected in the application programming interfaces (APIs) available for HDF5 and netCDF4. A number of HDF5's technical features are not supported by the netCDF data model, which on the other hand provides additional semantic features, most notably, shared dimensions and coordinates variables. The netCDF4 format and its predecessors are popular for the storage of Earth science datasets, including metocean ones. These formats allow the data to be placed into multidimensional arrays, called 'variables', in a hierarchical file system-like group structure. Arbitrary key-value metadata attributes can be attached to both groups and variables. The variables support various common data types, such as 1, 2, 4, and 8-byte integers, 2, 4, and 8-byte binary IEEE floating point numbers (Cowlishaw, 2008), and character strings. Also custom enumerations, variable-length arrays, and compound types can be defined, e.g., a combination of four floats and an integer. Furthermore, variables can be compressed transparently, i.e., without the user having to manually perform decompression before use.

Let us give a brief evaluation of support in software tools for the different file formats. Even if the delimiter-separated values files are not really standardized (however, see Lindner, 1993; Shafranovich, 2005), support for them is near universal. Software tools usually include options to deal with the particulars of the actual encoding (delimiter, quoting, headers, etc.), but this does require manual discovery of these specifics. These text-based formats can in principle be read and modified in a text-editor, but these are usually not designed to deal with large files, so this is actually impractical for all but the smallest datasets. The Excel 'xls' format, even though proprietary, has broad reading support. Support for HDF5 and netCDF4 formats in software tools is very extensive (The HDF Group, 2019b; Unidata, 2019b). This, next to their feature-set, is also a reason for us choosing to use them; they appear to be the most future-proof of the many binary formats in existence. We used Python modules to work with all these formats (McKinney et al., 2019; Colette, 2018; Unidata, 2019a).

Next let us consider the impact of a format being text-based or binary-based. Text-based formats in principle give a lot of freedom in choosing the format in which values are represented, but usually this is done in a single fixed-point format. To use the data, the values' representations need to be parsed into the standard binary number formats used by computers, namely, floats and integers of various kinds. Binary file formats use binary number formats directly, which are faster to load into memory and more space-efficient.[3] Because of their standardized nature, they can include other binary-specific features, such as transparent compression and checksums (data integrity codes).

Now let us look at the metadata. HDF5 and netCDF4 are considered self-describing formats, as they allow arbitrary metadata to be included next to the data. This data is easy to access, also programmatically. Table-based data files typically include one or two header lines of metadata (sometimes more), but there is no universal convention about what can be found there. So making use of information included in this way always requires user intervention. There are initiatives to create metadata inclusion standards for delimiter-separated values formats, but these have not gained significant adoption and are aimed at either web-based material (Tennison et al., 2015) or small datasets (Riede et al., 2010), or are very recent proposals (Walsh and Pollock, 2019).

Section 2.2.2 mentioned that the documentation available for the datasets we investigated is not machine-readable. It can be made so by providing it as metadata. Such metadata can be used to facilitate analyses and uses of the data. For example, if a tool has access to the range and units associated to series of values, air pressure and temperature, say, then it can automatically determine those for derived series, such as air density. Examples of metadata standards for datasets are the 'CF Conventions' (Eaton et al., 2017), ISO 19115-1 (ISO/TC 211, 2014), and the recently developed 'Metadata for wind energy Research & Development' (Sempreviva et al., 2017; Vasiljevic and Gancarski, 2019). It is encouraged in the Earth Science community to not just add arbitrary metadata, but include at least standard attributes from the 'CF Conventions' (Eaton et al., 2017) and follow the 'Attribute Convention for Data Discovery' (Earth Science Information Partners, 2015). These facilitate reuse, discovery, and also make it possible, for example, for software to enhance the presentation of the dataset elements (see, e.g., Hoyer et al., 2018). They also allow for adding further useful metadata, such as provenance information, e.g., in the form of an ISO Lineage (ISO/TC 211, 2019). These conventions are aimed at netCDF files, but can to a large degree by applied to HDF5 files as well. Of course the metadata to be included as recommended by these conventions can also be specified for table-based formats, but not in the same self-describing way.

## 3.2   Practical Experiences with Binary Formats

We already mentioned in Sect. 2.2.1 that we created binary HDF5 and netCDF4 file versions of the datasets we studied. In this section, we first, in Sect. 3.2.1, report on the process and its results. Then, in Sect. 3.2.2, we discuss the limitations of these formats, including limitations of software support.

---

[3]In text files, every decimal digit costs 8 bits (1 byte) to store, so a length-$n$ number requires $8n$ bits. In binary formats, a more efficient encoding is used (numbers as bit-strings), requiring $m$ bits. To round-trip from decimal to binary and back, $m = \lfloor n\log_2(10) + 2 \rfloor \approx \lfloor 3.3n + 2 \rfloor$ is sufficient (Matula, 1968). This picture does not change substantively if sign and magnitude are taken into account. In practice a 32-bit binary format is used for storing values, which uses 23 bits for representing the significand, sufficient for 6 decimal significant digits; 1 bit is used for the sign and 8 for the exponent.

### 3.2.1 The Transformation Process

Transforming the supplied data files was done by writing a specific script for each case. The general setup is similar for each script:

1. One needs to import the supplied datasets into in-memory data structures that can be manipulated by the scripting language. An important part of this step is the identification of missing data or data marked as faulty and encoding it appropriately. Storing them as the 'Not a Number' binary floating point value is the common approach we followed. Using a Boolean mask separate from the dataset itself is an alternative that can also be used in case the data stored does not consist of floating point values.

2. One must decide on and create a structure for the file, to organize the data and make it conveniently accessible. We used a hierarchical structure for this, grouping first by device (class) and then by quantity. For instrument locations, we tried two approaches:

   – adding the locations as groups in the hierarchy, below the 'quantity' groups (done for OWEZ);
   – collecting the data for all locations in a multidimensional array with additional axes next to the time axis, e.g., for height and boom direction (done for MMIJ and FINO1).

   For the different statistics (minimum, maximum, mean, and standard deviation), we tried three approaches:

   – adding the statistics series as separate variables in the hierarchy (done for all three);
   – keeping the statistics together in a compound data structure, essentially a tuple of values, where each value is accessed by (statistic) name; such compound values then formed the elements of the multidimensional arrays (done for MMIJ and FINO1);
   – adding the statistics as an extra axis to the multidimensional array (done as well for MMIJ).

3. One must collect and compose the metadata for the dataset, the devices, and the quantities. Then one must add these as attributes in the file. The latter is almost trivial to do once the former, time consuming task is completed.

4. One must choose an encoding and the storage parameters for the data and write it out to the file. We chose to store the values as 4-byte binary floating point numbers, compress it using the standard 'Deflate' algorithm, and add error detection using 'Fletcher-32' checksums. Furthermore, we used the information available about the accuracy of the values to round to the least significant *binary* digit. This is a lossy transformation that, however, does not lose significant information, but further improves compression.

Let us finish this section with some remarks.

– During the transformation process, we could load the datasets studied entirely into memory. This is convenient, but not necessary, as the process of reading the supplied datasets can be done in a piece-wise fashion.

- – The size of the files resulting from the transformation we made was one-eight of the supplied files' size or smaller and one-half their compressed size or smaller. (More precisely, the sizes of the uncompressed [compressed] supplied files versus the sizes of our HDF5 or netCDF4 versions are as follows. OWEZ: 1 GB [400 MB] vs. 65 MB; MMIJ: 500 MB [120 MB] vs. 55 MB; FINO1: 800 MB [120 MB] vs. 50 MB.)

- – Tools exist to facilitate the transformation process, most notably the on-line service Rosetta (Unidata, 2013), which generates netCDF files satisfying the CF Conventions.

- – Templates to facilitate the creation of netCDF files satisfying the CF Conventions and the Attribute Conventions for Data Discovery are available (NOAA National Centers for Environmental Information, 2015). These do not make use of hierarchical grouping, but can to a large degree be used within each group.

### 3.2.2 Limitations of Binary Formats Tested

When creating the transformed dataset files, we tested many of the features available in the HDF5 and netCDF4 formats. Not all of these features turned out to be as useful as initially expected or have sufficient software support. We here discuss features for which we encountered issues, to help others make an informed choice when considering their use.

**Compound data structures** Compound data structures are essentially tuples of values, where each component value is accessed by its name. These allow for a tight grouping of related data, for example to group all the statistics for a given signal for a given measuring interval, attach a quality flag, or to group the components of a vector (e.g., the wind velocity). However, metadata cannot be attached to the structure's components and to read any one component, the whole structure is loaded in memory, multiplying the memory requirements. Furthermore, support for creating these structures for use in netCDF4 files using Python was buggy and support for reading compound value data is currently far from universal; for example, it is not included in Matlab's netCDF interface. Also, documentation of their use is currently limited.

**2-byte floating point numbers** HDF5 allows storing 2-byte (16 bit) floating point numbers, which is more space-efficient if the precision is sufficient. The support in the core HDF5 library turned out to be buggy and support was non-existent, e.g., in Matlab.

**Scale-offset filters** Another approach for efficiently storing floating point values $x$ is to transform them to integer values $k$ of shorter bit-length. Namely choosing series-specific scale and offset parameters $\alpha$ and $\beta$ such that $x$ is equal to $\alpha k + \beta$ within required precision. HDF5 has a built-in filter to do this, but it does not preserve special floating point values like NaNs used for representing missing values. The 'CF Conventions' (Eaton et al., 2017) often used in netCDF files also describe a metadata-based approach, but not all software automatically applies the inverse transformation, so it is not transparent to the user.

**Dimensions** When creating variables, the netCDF4 format requires using defined 'dimensions' (e.g., time and height). These can be shared between variables and associated to 'coordinate variables' (e.g., arrays with concrete time values and instrument heights). There is also a similar concept of 'dimension scale' in HDF5, but it is not as convenient.

**Unicode** In principle both HDF5 and netCDF4 support Unicode text for group, variable, and attribute names and for attribute values. Software support for Unicode text in attribute values is not universal, however; notably, Matlab does not support this yet for netCDF4.

**String values** Both HDF5 and netCDF4 support variable-length strings as variable values. This can for example be useful for coordinate variables, such as when instrument position is designated by 'left' and 'right'. However, again Matlab does not support this yet for netCDF4.

## 4    Recommendations

Based on our analysis of the three datasets and on our work transforming them in to binary file formats, we have the following recommendations for the three main stakeholders. (We also briefly indicate their role in the shared responsibility for creating high-quality, well documented, and usable datasets.)

**Project owner** (Through the 'scope of work'-part of the contract with the dataset producer, this party can specify requirements for the dataset format, quality, and documentation, so that it meets the needs of the considered dataset users.)

- Require the dataset producer to provide the datasets in a standardized binary format.
- Agree with the dataset producers about a concrete level of quality control.
- Require the datasets to be accompanied by (explicitly specified) extensive metadata and documentation, including accuracy and quality information.

**Dataset producers** (Next to being responsible for producing the dataset, this party can inform the project owner about the possibilities for dataset creation and the dataset users about efficient dataset use.)

- Expand the automated checks performed on the signals the dataset series are based on, to efficiently remove avoidable issues that are currently still present (Sect. 2.2.1).
- Make the documentation of the dataset and its creation process more comprehensive (cf. Sect. 2.2.2). This is best done by attaching metadata right next to the data. External documentation such as data-manuals and websites, if still needed, can be semi-automatically generated from metadata that is stored in a structured way.
- Use clear version identification in dataset files, to avoid confusion when updated or extended datasets are released.
- Provide datasets in a binary format that allows for a structured combination of data and metadata (cf. Sect. 3.2). Based on our experience, we currently advise, for metocean measurement statistics datasets, using the netCDF4 format, with

- metadata added according to the Attribute Conventions for Dataset Discovery and CF Conventions (cf. Sect. 3.1),

- metadata describing absolute and relative sample uncertainty (cf. Sect. 2.2.5),

- coordinate variables for all dimensions of the data variables,

- each statistic series as a separate variable, so not using compound data structures or by expanding the multidimensional array,

- values binary-rounded according to the available uncertainties (cf. Sect. 2.2.5), which does not preclude inclusion of 'ancillary' variables for the uncertainty values themselves,

- sample standard deviations corrected for bias (cf. Sect. 2.2.5) or inclusion of an ancillary variable for the bias (modifying the values themselves may be seen as too invasive),

- variables compressed transparently, so not using a metadata-based scale-offset filter,

Its better support for dimensions and coordinate variables is what makes the netCDF4 format currently more attractive than the plain HDF5 format.

- Add a quality flag variable for each signal (cf. Sect. 2.2.4).

**Dataset users** (This party can communicate its needs and provide feedback to to the project owner and dataset producers.)

- Invest in learning to work with format like HDF5 or netCDF4, as this will allow working more efficiently with datasets (cf. Sect. 3).

- Provide feedback to the dataset producers about issues encountered and dataset features that would have added value for research (our experience in this regard is positive).

- And of course do not trust the data blindly and perform some checks in the vein of those we discussed in Sect. 2.2.1.

## 5 Conclusions

The questions of our study were: (i) Are these issues commonly shared in metocean measurement datasets? (ii) How can the issues that are present be addressed?

The answer to the first question is 'yes, but not uniformly': The analysis of three datasets with statistics of metocean signals aimed at wind energy applications presented in Sec. 2.2 showed that indeed there are shared issues, such as the presence of unmarked faulty data (outliers, most clearly), incomplete documentation (signal accuracy, most generally), and value encoding (lack of uncertainty information, most importantly). Some issues are not shared, and one dataset can actually be seen as an example of good practice in some aspect (the quality flags included in the FINO1 dataset, most concretely).

An abstract answer to the second question is 'by the dataset producers, in a straightforward way, with limited effort'. More concretely:

- The techniques we used to bring faulty data to light are straightforward to implement, which supports our claim that they can be detected and fixed with relatively little effort.

- Concerning documentation: In our quest for creating a good overview of the datasets, we collected information from various sources to supplement the documentation provided; this is a time consuming task. Much of the information that we had to search for, is available to the datasets producers, so the effort for them is smaller. Given that one cannot expect all dataset users to perform data quality analyses and information collection efforts themselves, it would be beneficial if the project owners explicitly make this a duty of the dataset producers. This will make their datasets more useful and therefore more valuable.

- As noted above, a specific issue with the datasets was the limited information about and quantification of the uncertainty of the dataset values. The expressions for uncertainties and bias we derived provide a straightforward quantification of the statistics' uncertainties and bias based on the information that is typically available, absolute and relative uncertainties for the sample values. These expressions can be used by users if needed by their application. The dataset producers can also apply them and use the uncertainty values found to improve their dataset, e.g., by rounding the dataset values (reducing the size requirements) or by including the uncertainty values as ancillary variables.

- In support of our analysis of the datasets, we created versions in a binary format. In comparison to the tabular formats in which the datasets are made available, such binary formats are more convenient for users, as they make the data available in a much more structured format and as they are self-describing when documentation is added as metadata. The description of our effort, experiences, and feature evaluation provide a high-level guide and suggested best practices to dataset producers who wish to also improve their datasets in this way.

In summary, *this paper shows why and how metocean measurement datasets for wind energy applications can be improved in various, useful ways, with relatively little effort.* This effort can be seen by the project owner as necessary for getting the most value out of the raw data collected. Such a well-documented dataset with uncertainty and quality information included creates the possibility for consciously making possibly different choices (trade-offs) when setting up future measurement campaigns.

*Code and data availability.* Code used during the research is publicly available via GitHub and Zenodo (Quaeghebeur, 2020). This bundle also includes the metadata included in the transformed datasets as human-readable and machine-readable YAML files.

We are not allowed to make the transformed FINO1 dataset available. It is not yet clear whether we will obtain permission to make the transformed OWEZ and MMIJ datasets available. If we do, these will be put on a publicly available data repository, referenced in an updated version of the bundle (Quaeghebeur, 2020).

**Appendix A: The Datasets and Their Analysis**

**A1 A First Look at the Datasets**

**A1.1 MMIJ — Measuring Mast IJmuiden**

In the context of a Dutch governmental research program, a met mast was built in the Dutch part of the North Sea with the
5 aim to gather metocean data with a frequency and quality needed for the planning and development of offshore wind farms
in the Dutch North Sea. Its location is $52°50'53.4''$ North, $3°26'8.4''$ East (WGS 84), which is 82 km off the Dutch coast
near the province North-Holland. The location is indicated in Fig. 1. The mast was ready for operation in 2011 and was
decommissioned by 2017. Data is available for the period November 2011–March 2016. Multiple datasets can be obtained;
we restricted attention to the one for meteorological signals. The instruments used and quantities measured, and some of their
10 characteristics are listed in Table A1.

The MMIJ datasets can be obtained by registering, which is free, and filling in a request form on a website of the Energy
research Centre of the Netherlands (ECN, 2019).[4] The meteorological statistics dataset can be downloaded via an e-mailed link
as a single compressed semicolon-separated values (csv) file. The total size is a good 500 MB, or about 120 MB compressed.
This represents data points for 229 248 10-minute intervals. The data in the csv file is structured as follows:

15  – 1 date-time column (`YYYY-MM-DD hh:mm`);

  – 65 sets of four columns each: one for each of the four real-valued statistics, 'min', 'max', 'avg', and 'std'; each set
    corresponding to a specific measured quantity and location on the mast.

The statistics' values are encoded in a decimal fixed-point format with five fractional digits (`x...x.xxxxx`).

Information about the dataset, the met mast, and its context is available through the same website. In particular, there is
20 an instrumentation report (Werkhoven and Verhoef, 2012). Some information about the instruments used and in particular the
measurement uncertainty had to be looked up in specification sheets. Further clarifications were obtained through personal
communication with people involved in the project (cf. Acknowledgements).

**A1.2 FINO1 — Research Platform in the North Sea and the Baltic Sea Nr. 1**

In the context of the German governmental research program FINO (for 'Forschungsplattformen in Nord- und Ostsee') started
25 in 2002, three measuring stations with met masts were built; two in the German part of the North Sea and one in the Baltic. The
aim is supporting technological developments for and study the effect of offshore wind farms. We have looked at data from
the first mast erected, FINO1, which became operational in 2003. Its location is $54°0'53.5''$ North, $6°35'15.5''$ East (WGS 84),
45 km North of the island of Borkum, near the site where the offshore wind farm 'Alpha Ventus' was built in 2009–2010. The
location is indicated in Fig. 1. Data from 2004 onward is available; measurements are still ongoing. Multiple datasets can be

---

[4]Since the work reported on in this paper was carried out, ECN has become part of TNO, the Netherlands Organisation for applied scientific research. Its
name will change in the coming period.

**Table A1.** An overview of the instruments and their locations on the MMIJ met mast (height in meters above Lowest Astronomical Tide), the quantity measured, measurement uncertainty, the measurement ranges, and the sampling frequencies.

| Instrument (#) | Heights [m] | Pos.[b] | Quantity[qc] | Unit | Uncertainty[m] abs. | rel. [%] | Range[m] | Freq. [Hz] |
|---|---|---|---|---|---|---|---|---|
| cup anemometer (8) | 27, 58.5 92 | reg. irr. | hor. wind sp.[1] | $\mathrm{m\,s^{-1}}$ | 0.2 | 1 | 0.3–75 | 4 |
| ultrasonic anemometer (3) | 85 | reg. | status[1] | – | | | $\{-10^3, 0\}$[eo] | 4 |
| | | | wind sp. X dir.[1] | | | | | |
| | | | wind sp. Y dir.[1] | $\mathrm{m\,s^{-1}}$ | 0.1 | 2 | −60–60 | 4 |
| | | | wind sp. Z dir.[1] | | | | | |
| wind vane (9) | 27, 58.5, 87 | reg. | wind direction[1] | ° | 1 | | 0–360[d] | 4 |
| barometer (2) | 21, 90 | | atm. pressure[1] | hPa | 0.1 | | 500–1100 | 4 |
| thermometer[i] (2) | 21, 90 | | ambient temp.[1] | °C | 0.12 | | −80–60 | 4 |
| hygrometer[i] (2) | 21, 90 | | rel. humidity[1] | % | 1 | | 0–100 | 4 |
| precipitation detector (1) | 27 | U | precip. presence[1] | – | | | $\{0, 100\}$[en] | 4 |
| precipitation monitor (2) | 21 | l, r | status[1] | – | | | $\{0, 100\}$[eo] | 4 |
| | | | quality[5] | % | | | 0–100 | 4 |
| | | | synoptic code[5] | – | | | $\{0, \ldots, 99\}$[ew] | 4 |
| | | | precip. presence[5] | – | | | $\{0, 100\}$[en] | 4 |
| | | | precip. intensity[5] | $\mathrm{mm\,min^{-1}}$ | | 15 | 0.005–250 | 4 |
| | | | precip. amount[5,r] | mm | | | 0– | 4 |
| | | | visibility[5] | m | | | 0–10 000 | 4 |
| from[v,s] cup anemometer (3) | 27, 58.5, 92 | | hor. wind sp.[1] | $\mathrm{m\,s^{-1}}$ | 0.14 | | 0.3–75 | 4 |
| from[v] ultrasonic anemometer (3) | 85 | reg. | wind sp. magn.[1] | $\mathrm{m\,s^{-1}}$ | 0.07 | | 0–104 | 4 |
| | | | hor. wind sp.[1] | $\mathrm{m\,s^{-1}}$ | 0.07 | | 0–85 | 4 |
| from[v,s] ultrasonic anemometer (1) | 85 | | hor. wind sp.[1] | $\mathrm{m\,s^{-1}}$ | 0.05 | | 0–85 | 4 |
| from[v,s] wind vane (3) | 27, 58.5, 87 | | wind direction[1] | ° | 0.7 | | 0–360[d] | 4 |
| from[v] barom. and thermom. (2) | 21, 90 | | air density[1] | $\mathrm{kg\,m^{-3}}$ | 0.0001 | | 0.5–2.0 | 4 |

[b] For instruments on booms, positions are boom orientations [°], with (geographic) North at 46.5°, 'reg.' corresponds to $\{0, 120, 240\}$ and 'irr.' to $\{180, 300\}$. For those not on booms other identifiers are used, if known.

[d] Means lie between 0°–360°; minima and maxima can be outside of that interval so that min ≤ avg ≤ max.

[en] No, Yes.    [eo] '0' = OK, non-zero = Not OK.    [ew] Using synoptic 'present weather' codes defined by the World Meteorological Organization (2016, p. 356–358).

[i] Thermometer and hygrometer are contained in a single package.    [m] Missing values are unknown.

[qc] Quality code: '1' = 'ISO 17025 approved, in accordance with IEC61400-12'; '5' = 'no or unknown calibration'.

[r] Between sensor resets.    [s] Correction for tower shadow by selective averaging of values at the same height.

[v] Virtual measurement; namely, derived from signals obtained with one or more actual instruments.

**Table A2.** An overview of the instruments and their locations on the FINO1 met mast (height in meters above Lowest Astronomical Tide), the quantity measured, measurement uncertainty, the measurement ranges, and the sampling frequencies.

| Instrument (#) | Heights [m] | Quantity | statistics[s] | Unit | Uncertainty[m] abs. | rel. [%] | Range[m] | Freq. [Hz] |
|---|---|---|---|---|---|---|---|---|
| cup anemometer (8) | $34, 41, 51, 61,$ $71, 81, 91, 102$ | hor. wind sp. | $- + \mu\,\sigma$ | $\mathrm{m\,s^{-1}}$ | 0.1 | 1 | 0.1–75 | 1 |
| ultrasonic anemometer (3) | $42, 62, 82$ | hor. wind sp. | $- + \mu\,\sigma$ | $\mathrm{m\,s^{-1}}$ | 0.01 | 1 | 0–45 | 50 |
| | | wind direction | $\mu\,\sigma$ | ° | | 1 | 0–359 | 50 |
| wind vane (9) | $34, 51, 71, 91$ | wind direction | $+\mu\,\sigma$ | ° | | 2 | 0–360 | 1 |
| barometer (2) | $21, 93$ | atm. pressure | $\mu$ | hPa | 0.3 | | 800–1060 | 1 |
| thermometer[i] (5) | $34, 42, 52, 72, 101$ | ambient temp. | $\mu$ | °C | 0.1 | | | 1 |
| hygrometer[i] (5) | $34, 42, 52, 72, 101$ | rel. humidity | $\mu$ | % | 3 | | 10–100 | 1 |
| precipitation monitor (2) | $23, 101$ | precip. presence | meas.[v] | – | | | $\{0,1\}^{\mathrm{c}}$ | |
| precipitation sensor (1) | 23 | precip. intensity | $\mu$ | mA | | | 4–20 | 1 |
| pyranometer (2) | $34, 93$ | global radiation | $\mu$ | $\mathrm{W\,m^{-2}}$ | | 3 | 0–4000 | 1 |

[c] No, Yes.   [m] Missing values are unknown.   [i] Thermometer and hygrometer are contained in a single package.

[s] Statistics included (with column name): '$-$' = minimum ('Minimum'), '$+$' = maximum ('Maximum'), '$\mu$' = mean ('Value'), '$\sigma$' = standard deviation ('Deviation').

[v] The measurement is given (in the 'Value' column), as there is essentially one measurement per ten minutes.

obtained; again we restricted attention to the one for meteorological signals. The instruments used and quantities measured, and some of their characteristics are listed in Table A2.

The FINO1 datasets can be obtained after requesting access (BSH, 2019a), which is free for academic research, but not so for commercial purposes; re-dissemination is not allowed. Credentials are then provided to login to the download website (BSH, 2019a), where one can select the desired signals and time period. The resulting dataset is delivered as a compressed set of tab-separated values (dat) files, one for each selected quantity/height combination. We selected the meteorological statistics data for the years 2004–2016. The total size is a good 800 MB, or about 120 MB compressed. This represents data points for 683 856 10-minute intervals. The data in each dat file is structured as follows:

- 1 date-time column (`YYYY-MM-DD hh:mm:ss`);

- 4 statistics columns, 'Value', 'Minimum', 'Maximum', and 'Deviation';

- 1 quality column ('0' = raw, '1' = doubtful quality, '2' = quality controlled).

The statistics' values are encoded in a decimal fixed-point format with up to two fractional digits (`x...x.xx`).

Information about the dataset, the met mast, and its context is available through the platform's websites (FINO 1, 2019; BSH, 2019b). A detailed overview table regarding the mast's instrumentation (DEWI, 2015) is available upon request by email. Some

information about the instruments used and in particular the measurement ranges had to be looked up in specification sheets. Further clarifications were obtained through personal communication with people involved in the project (cf. Acknowledgements).

Others have looked at the FINO1 data before. For example, an initial data analysis was presented after five years of operation (Beeken et al., 2009) and detailed studies have been performed on the wind speed data gathered (Westerhellweg et al., 2012; Stepek et al., 2015).

### A1.3 FAIRness analysis

There is currently a movement in the academic community to try and make datasets FAIR: findable, accessible, interoperable, and reusable (Wilkinson et al., 2016). This appendix provides a brief analysis of the FAIRness of the three datasets that were investigated. It first looks at the current status, then moves to what role the recommendations of this paper play in changing that status, to finally evaluate the role of the non-user stakeholders. Our analysis is based on the checklist "How FAIR is your data?" of Jones and Grootveld (2017).

We look at each of the FAIRness principles:

**Findable** None of the datasets has a persistent identifier assigned to it. While metadata for each dataset is available online, it is for none of the datasets present in a searchable resource, but less conveniently in manuals or on a custom website. So none of the datasets are really findable (according to the FAIRness criteria).

**Accessible** For all of the datasets, the protocol by which the data can be retrieved follows an recognized standard; namely, it can be downloaded from a website. Furthermore, even if obtaining the data requires authorisation as for MMIJ and FINO1, the available metadata is accessible without it. So, setting aside the lack of a persistent identifier, all the datasets are quite accessible (according to the FAIRness criteria).

**Interoperable** All of the datasets are provided in a commonly understood format, although the format for OWEZ (old, proprietary Excel format) is not open. The metadata provided does not follow any standard and neither are controlled vocabularies used. Also, no qualified references or links to other (meta)data are provided. Given the above, all the datasets are only interoperable in a very basic way (according to the FAIRness criteria).

**Reusable** The (meta)data are fairly accurate and reasonably well described for all three datasets. Only FINO1 has a fairly clear (but restrictive) license. For all datasets, the provenance is clear. While collected for wind energy applications, the datasets contain Earth Science data; the metadata standards relevant in that domain are not met. Based on the above, the datasets are somewhat reusable (according to the FAIRness criteria).

This paper's recommendations argue for the data to be made available in a standardized binary format with metadata included. It also promotes more extensive quality checks. Such transformed datasets would raise the level of interoperability and reusability, mostly because of the improved handling of metadata.

The dataset producers can furthermore make sure the datasets are assigned a persistent identifier pointing to a location in a data repository, where they are stored under a clear license. They could make the metadata collected for inclusion in the binary dataset file also available there. These efforts would raise the level of findability, accessibility, and reusability. In line with what was mentioned in our recommendations (Sect. 4), the project owner can specify the FAIRness criteria as requirements, to

ensure that this is actually done.

## A2   Dataset Issues

### A2.1   Maximum and Minimum for Directional Data

We here give a proposal for definitions of maximum and minimum for directional data. We assume the sampling frequency is high enough to make direction changes larger than $180°$ for successive samples practically impossible.

Transform the direction sequence from $0°$–$360°$ to the real line so that '$360°$ jumps' are removed; e.g., the sequence $356°$, $358°$, $1°$, and $4°$ would become $356°$, $358°$, $361°$, and $364°$. Call the minimum and maximum of this transformed sequence $\chi$ and $\xi$; so $\chi = 356°$ and $\xi = 364°$ in our example. If $\xi - \chi > 360°$ the direction has changed at least one full rotation for the given sequence. Let $\mu$ be the (vector) mean, expressed within $0°$–$360°$; so $\mu \approx 359.75°$ in our example. Now choose $k$ such that $\chi + k\,360° \le \mu \le \xi + k\,360°$ with $\max\{|\chi + k\,360° - \mu|, |\xi + k\,360° - \mu|\}$ minimal; $k = 0$ in our example. Then $\chi + k\,360°$

and $\xi + k\,360°$ are the sought for minimum and maximum.

### A2.2   Statistic Value Uncertainty

The statistics present in the dataset are derived from $n$ measurements $x_k$ uniformly sampled over a length-$T$ interval, where $T = 600$ s for the datasets we consider. To get a view on the uncertainty of the statistics, we model the process generating the measurements as follows: There is an underlying signal $y$ with samples $y_k = y(t_k)$. On measurement, noise is added, so that

$x_k = y_k + e_k$ for all $k \in \{1, \dots, n\}$. The noise is assumed to consist of independent absolute and relative zero-mean Gaussian components (Cramér, 1946, Chapter 17), i.e., $e_k = \varepsilon_a z_{a,k} + \varepsilon_r y_k z_{r,k}$ with $z_{r,k}$ and $z_{a,k}$ samples from independent standard normal distributions, so that the component's standard deviations are $\varepsilon_a$ and $\varepsilon_r y_k$.

We first consider the contribution of sampling and then the contribution of the noise to the uncertainty of the statistics.

**Uncertainty due to sampling**

The 'ideal' statistic values are defined in terms of the continuous-time signal:

$$\check{y}_c = \min_{t \in [0,T]} y(t), \qquad \hat{y}_c = \max_{t \in [0,T]} y(t), \qquad \bar{y}_c = \frac{1}{T}\int_0^T y(t)\,\mathrm{d}t, \qquad s_{y,c}^2 = \frac{1}{T}\int_0^T (y(t) - \bar{y}_c)^2\,\mathrm{d}t. \tag{A1}$$

The 'noiseless' sample statistics values are

$$\check{y} = \min_{k \in \{1,\dots,n\}} y_k, \qquad \hat{y} = \max_{k \in \{1,\dots,n\}} y_k, \qquad \bar{y} = \frac{1}{n}\sum_{k=1}^n y_k, \qquad s_y^2 = \frac{1}{n}\sum_{k=1}^n (y_k - \bar{y})^2, \tag{A2}$$

where for the sample variance $s_y^2$, we did not apply the usual bias correction because $n$ is assumed sufficiently large.

As we assume is done in the datasets, we take $t_k = (k-1)\frac{1}{n}T$. So we are applying the 'Left Rule' numerical integration method (see, e.g., Tucker, 1997) to get estimates $\bar{y}$ for $\bar{y}_c$ and $s_y^2$ for $s_{y,c}^2$. A corresponding error estimate is $\frac{T^2}{2n}\left|\sum_{k=1}^n f'(t_k)\right|$, where $f$ is equal to $\frac{1}{T}y$ and $\frac{1}{T}(y-\bar{y}_c)^2$ respectively. An estimate for the sum of derivatives is obtained by assuming $y$ is linear, i.e., $y' \approx \frac{\hat{y}-\check{y}}{T}$ and $\left((y-\bar{y}_c)^2\right)' = 2(y-\bar{y}_c)y' \approx 2s_y\frac{\hat{y}-\check{y}}{T}$. Similarly, for uncertainty estimates of the maximum and minimum statistics we assume that the signal continues to linearly increase (decrease) for half a sample step beyond the maximum (minimum) sample.

To get concrete values, we replace the noiseless statistics with the actual noisy ones. This results in the following expressions:

$$\tau_{\check{y}} \approx \frac{\hat{x}-\check{x}}{2n}, \qquad \tau_{\hat{y}} \approx \frac{\hat{x}-\check{x}}{2n}, \qquad \tau_{\bar{y}} \approx \frac{\hat{x}-\check{x}}{2n}, \qquad \tau_{s_y^2} \approx s_x\frac{\hat{x}-\check{x}}{n}, \qquad \tau_{s_y} \approx \frac{1}{2s_x}\tau_{s_y^2} \approx \frac{\hat{x}-\check{x}}{2n}, \qquad \text{(A3)}$$

where the uncertainty for the standard deviation $s_y$ was derived from the one for the variance by applying a first order Taylor approximation of the square root. In case the minimum and maximum statistics are not available, but the sample standard deviation is, one could use the crude estimates $\check{x} \approx \bar{x} - z_{1-1/n}s_x$ and $\hat{x} \approx \bar{x} + z_{1-1/n}s_x$, where $z_{1-1/n}$ is the standard normal quantile for exceedance probability $1/n$.

### A2.3 Uncertainty due to measurement noise

We use the following random variables to model the process that adds noise to the measurements: $X_k$ for the measurements and $E_k$ for the noise, with auxiliary standard normal variables $Z_{a,k}$ and $Z_{r,k}$, so that $X_k = y_k + E_k$ with $E_k = \varepsilon_a Z_{a,k} + \varepsilon_r y_k Z_{r,k}$. Here, the basic random variables $Z_{a,k}$ and $Z_{r,k}$ are assumed to be independent from each other and all other random variables $Z_{a,\ell}, Z_{r,\ell}, \ell \neq k$.

Some further notation: $\mathbb{E}$ is the expectation operator. Var and Cov are the variance and covariance operators, respectively, defined by for any random variables $V$ and $W$ by $\text{Cov}(V,W) = \mathbb{E}\big((V-\mathbb{E}(V))(W-\mathbb{E}(W))\big)$ and $\text{Var}(V) = \text{Cov}(V,V)$. Furthermore, we let $\check{V} = \min_{k=1}^n V_k$, $\hat{V} = \max_{k=1}^n V_k$, $\bar{V}^{(p)} = \frac{1}{n}\sum_{k=1}^n V_k^p$, with $\bar{V} = \bar{V}^{(1)}$, and $s_V^2 = \frac{1}{n}\sum_{k=1}^n(V_k-\bar{V})^2$.

Recall that standard normal variables $Z$ are completely determined by their expectation $\mathbb{E}(Z) = 0$ and variance $\text{Var}(Z) = \mathbb{E}(Z^2) = 1$. Also, the expectation of any odd power is zero: $\mathbb{E}(Z^{2m+1}) = 0$ (Cramér, 1946, Equation 17.2.3).

For the sample minimum and maximum we assume that the measurement noise does not substantially influence the order statistics, so $\check{X} = \check{y}+E_{\check{k}}$ and $\hat{X} = \hat{y}+E_{\hat{k}}$. (Otherwise this noise introduces bias in the estimate and an extra term in the variance (see Cramér, 1946, Equation 28.6.16).) This implies

$$\check{x} \approx \mathbb{E}(\check{X}) = \check{y} + \mathbb{E}(E_{\check{k}}) = \check{y}, \qquad\qquad \sigma_{\check{y}}^2 = \text{Var}(\check{X}) = \text{Var}(E_{\check{k}}) = \varepsilon_a^2 + \varepsilon_r^2\check{y}^2, \qquad\qquad \text{(A4)}$$

$$\hat{x} \approx \mathbb{E}(\hat{X}) = \hat{y} + \mathbb{E}(E_{\hat{k}}) = \hat{y}, \qquad\qquad \sigma_{\hat{y}}^2 = \text{Var}(\hat{X}) = \text{Var}(E_{\hat{k}}) = \varepsilon_a^2 + \varepsilon_r^2\hat{y}^2, \qquad\qquad \text{(A5)}$$

because

$$\mathbb{E}(E_k) = \varepsilon_a\mathbb{E}(Z_{a,k}) + \varepsilon_r y_k\mathbb{E}(Z_{r,k}) = 0,$$

$$\text{Var}(E_k) = \varepsilon_a^2\,\text{Var}(Z_{a,k}) + \varepsilon_r^2\check{y}^2\,\text{Var}(Z_{r,k}) = \varepsilon_a^2 + \varepsilon_r^2\check{y}^2,$$

where for the variance the first equality follows from independence of the variables $Z_{a,k}$ and $Z_{r,k}$.

For the sample mean we can deduce that

$$\bar{x} \approx \mathbb{E}(\bar{X}) = \bar{y} + \mathbb{E}(\bar{E}) = \bar{y} \qquad \text{and} \qquad \sigma_{\bar{y}}^2 = \mathrm{Var}(\bar{X}) = \mathrm{Var}(\bar{E}) = \frac{1}{n}\left(\varepsilon_a^2 + \varepsilon_r^2(\bar{y}^2 + s_y^2)\right) \tag{A6}$$

because

$$\mathbb{E}(\bar{E}) = \frac{1}{n}\sum_{k=1}^{n}\left(\varepsilon_a \mathbb{E}(Z_{a,k}) + \varepsilon_r \mathbb{E}(Z_{r,k})\right) = 0,$$

$$\mathrm{Var}(\bar{E}) = \frac{1}{n^2}\sum_{k=1}^{n}\left(\varepsilon_a^2\,\mathrm{Var}(Z_{a,k}) + \varepsilon_r^2 y_k^2\,\mathrm{Var}(Z_{r,k})\right)$$

$$= \frac{1}{n^2}\sum_{k=1}^{n}(\varepsilon_a^2 + \varepsilon_r^2 y_k^2) = \frac{1}{n}(\varepsilon_a^2 + \varepsilon_r^2 \frac{1}{n}\sum_{k=1}^{n} y_k^2) = \frac{1}{n}(\varepsilon_a^2 + \varepsilon_r^2 \bar{y}^{(2)}) = \frac{1}{n}(\varepsilon_a^2 + \varepsilon_r^2(\bar{y}^2 + s_y^2)),$$

because it holds that $\bar{y}^{(2)} = \bar{y}^2 + s_y^2$ (Cramér, 1946, Equation 15.4.4).

For the sample standard deviation $s_X$, we use the first order Taylor expansion of the square root $s_X = \sqrt{s_X^2}$ with $s_X^2$ varying

10  around $\mathbb{E}(s_X^2)$:

$$\sqrt{s_X^2} \approx \sqrt{\mathbb{E}(s_X^2)} + \frac{1}{2\sqrt{\mathbb{E}(s_X^2)}}\left(s_X^2 - \mathbb{E}(s_X^2)\right).$$

So first order approximations of the expectation and variance are

$$s_x \approx \mathbb{E}(s_X) \approx \sqrt{\mathbb{E}(s_X^2)} \qquad \text{and} \qquad \sigma_{s_y}^2 = \mathrm{Var}(s_X) \approx \frac{1}{4\mathbb{E}(s_X^2)}\,\mathrm{Var}(s_X^2). \tag{A7}$$

So we see that we actually need to calculate $\mathbb{E}(s_X^2)$ and $\mathrm{Var}(s_X^2)$, the expectation and variance of the sample variance.

15  Let us first write this sample variance in terms of our model variables:

$$s_X^2 = \bar{X}^{(2)} - \bar{X}^2 = \bar{y}^{(2)} + 2\overline{yE} + \bar{E}^{(2)} - \bar{y}^2 - 2\bar{y}\bar{E} - \bar{E}^2 = s_y^2 + s_E^2 + 2(\overline{yE} - \bar{y}\bar{E}).$$

Then

$$\mathbb{E}(s_X^2) = s_y^2 + \mathbb{E}(s_E^2) + 2\left(\mathbb{E}(\overline{yE}) - \bar{y}\mathbb{E}(\bar{E})\right) = s_y^2 + \mathbb{E}(s_E^2) + 0 = s_y^2 + \mathbb{E}(\bar{E}^{(2)} - \bar{E}^2) = s_y^2 + (1 - \frac{1}{n})(\varepsilon_a^2 + \varepsilon_r^2 \bar{y}^{(2)}) \tag{A8}$$

because

$$\mathbb{E}(\overline{yE}) = \frac{1}{n}\sum_{k=1}^{n} y_k \mathbb{E}(E_k) = 0,$$

$$\mathbb{E}(\bar{E}^{(2)}) = \frac{1}{n}\sum_{k=1}^{n}\left(\varepsilon_a^2 \mathbb{E}(Z_{a,k}^2) + 2\varepsilon_a\varepsilon_r y_k \mathbb{E}(Z_{a,k})\mathbb{E}(Z_{r,k}) + \varepsilon_r^2 y_k^2 \mathbb{E}(Z_{r,k}^2)\right) = \varepsilon_a^2 + \varepsilon_r^2 \bar{y}^{(2)},$$

$$\mathbb{E}(\bar{E}^2) = \frac{1}{n^2}\sum_{k=1}^{n}\sum_{\ell=1}^{n}\left(\varepsilon_a^2 \mathbb{E}(Z_{a,k}Z_{a,\ell}) + \varepsilon_a\varepsilon_r\left(y_k \mathbb{E}(Z_{a,\ell})\mathbb{E}(Z_{r,k}) + y_\ell \mathbb{E}(Z_{a,k})\mathbb{E}(Z_{r,\ell})\right) + \varepsilon_r^2 y_k y_\ell \mathbb{E}(Z_{r,k}Z_{r,\ell})\right)$$

$$= \frac{1}{n}(\varepsilon_a^2 + \varepsilon_r^2 \bar{y}^{(2)}).$$

Furthermore

$$\mathrm{Var}(s_X^2) = \mathrm{Var}(s_E^2) + 4\,\mathrm{Var}(y\bar{E} - \bar{y}\bar{E}) + 4\,\mathrm{Cov}(s_X^2, y\bar{E} - \bar{y}\bar{E}).$$

The last term of this expression is zero because all terms of its expansion contain odd powers of independent standard normal random variables. We do not perform the tedious calculation of the first term, as it essentially expresses the uncertainty of the measurement noise, which has been left unmodeled. Therefore we *ignore* this term, which means we consider a lower bound:

$$
\begin{aligned}
\frac{1}{4}\,\mathrm{Var}(s_X^2) &\geq \mathrm{Var}(y\bar{E} - \bar{y}\bar{E}) \\
&= \mathrm{Var}(y\bar{E}) + \bar{y}^2\,\mathrm{Var}(\bar{E}) - 2\bar{y}\,\mathrm{Cov}(y\bar{E}, \bar{E}) \\
&= \mathbb{E}(y\bar{E}^2) - \mathbb{E}(y\bar{E})^2 + \bar{y}^2\big(\mathbb{E}(\bar{E}^2) - \mathbb{E}(\bar{E})^2\big) - 2\bar{y}\big(\mathbb{E}(y\bar{E}\bar{E}) - \mathbb{E}(y\bar{E})\mathbb{E}(\bar{E})\big) \\
&= \mathbb{E}(y\bar{E}^2) + \bar{y}^2\mathbb{E}(\bar{E}^2) - 2\bar{y}\mathbb{E}(y\bar{E}\bar{E}) \\
&= \frac{1}{n}\big(\varepsilon_\mathrm{a}^2\bar{y}^{(2)} + \varepsilon_\mathrm{r}^2\bar{y}^{(4)}\big) + \bar{y}^2\frac{1}{n}\big(\varepsilon_\mathrm{a}^2 + \varepsilon_\mathrm{r}^2\bar{y}^{(2)}\big) - 2\bar{y}\frac{1}{n}\big(\varepsilon_\mathrm{a}^2\bar{y} + \varepsilon_\mathrm{r}^2\bar{y}^{(3)}\big),
\end{aligned}
$$

where in the last step the first and last terms' calculation is analogous to the one of $\mathbb{E}(\bar{E}^2)$ above. It holds that $\bar{y}^{(2)} = \bar{y}^2 + s_y^2$ and because we have no estimate for $\bar{y}^{(3)}$ and $\bar{y}^{(4)}$, we use the Gaussian case, i.e., we assume $\bar{y}^{(3)} \approx \bar{y}^3 + 3\bar{y}s_y^2$ and $\bar{y}^{(4)} \approx \bar{y}^4 + 6\bar{y}^2 s_y^2 + 3s_y^4$ (Johnson et al., 1994, Ch. 13). This gives

$$\frac{n}{4}\,\mathrm{Var}(s_X^2) \geq \varepsilon_\mathrm{a}^2 s_y^2 + \varepsilon_\mathrm{r}^2\big(\bar{y}^{(4)} + \bar{y}^2\bar{y}^{(2)} - 2\bar{y}\bar{y}^{(3)}\big) \approx \varepsilon_\mathrm{a}^2 s_y^2 + \varepsilon_\mathrm{r}^2 s_y^2(\bar{y}^2 + 3s_y^2) = s_y^2\big(\varepsilon_\mathrm{a}^2 + \varepsilon_\mathrm{r}^2(\bar{y}^2 + 3s_y^2)\big).$$

Going back to the sample standard deviation in Eq. A7, using Eq. A8, and assuming $n \gg 1$ and $\varepsilon_\mathrm{r}^2 \ll 1$ we get

$$s_x \approx \mathbb{E}(s_X) \approx \sqrt{s_y^2 + (\varepsilon_\mathrm{a}^2 + \varepsilon_\mathrm{r}^2\bar{y}^{(2)})} \qquad \text{so} \quad s_y^2 \approx \frac{1}{1 + \varepsilon_\mathrm{r}^2}\big(s_x^2 - (\varepsilon_\mathrm{a}^2 + \varepsilon_\mathrm{r}^2\bar{y}^2)\big) \approx s_x^2 - (\varepsilon_\mathrm{a}^2 + \varepsilon_\mathrm{r}^2\bar{y}^2), \tag{A9}$$

$$\sigma_{s_y}^2 = \mathrm{Var}(s_X) \geq \frac{s_y^2}{s_y^2 + (\varepsilon_\mathrm{a}^2 + \varepsilon_\mathrm{r}^2\bar{y}^{(2)})}\frac{1}{n}\big(\varepsilon_\mathrm{a}^2 + \varepsilon_\mathrm{r}^2(\bar{y}^2 + 3s_y^2)\big) \approx \frac{1}{n}\big(\varepsilon_\mathrm{a}^2 + \varepsilon_\mathrm{r}^2(\bar{y}^2 + 3s_y^2)\big), \tag{A10}$$

where for the last approximation we assumed that the measurement noise's contribution to the sample standard deviation is negligible ($s_y^2 \gg \varepsilon_\mathrm{a}^2 + \varepsilon_\mathrm{r}^2\bar{y}^2$).[5] In any case, in general the bias in $s_x$ as an estimator of $s_y$ dwarfs the estimate of the uncertainty $\sigma_{s_y}$ due to the measurement noise. Even the uncertainty in the bias (the unmodeled uncertainty of the measurement noise) may overwhelm $\sigma_{s_y}$. These considerations lead us to conclude that the lower bound we give is conservative in general and that the real uncertainty can be substantially larger.

To get concrete values, we replace $\check{y}$, $\hat{y}$, $\bar{y}$ and $s_y^2$ appearing in the expressions for the uncertainties by their estimates. We also deal with the corner case $s_x^2 < \varepsilon_\mathrm{a}^2 + \varepsilon_\mathrm{r}^2\bar{x}^2$. This results in the following estimates for the expectations and uncertainty, again

---

[5]The standard for what is negligible differs between estimates of statistics and of uncertainties thereof. For example, $\frac{\varepsilon_\mathrm{a}^2 + \varepsilon_\mathrm{r}^2\bar{y}^2}{s_y^2} = 10\%$ is non-negligible in Eq. A9, but is negligible in Eq. A10.

assuming $\varepsilon_\mathrm{r}^2 \ll 1$:

$$\check{y} \approx \check{x}, \qquad\qquad\qquad\qquad\qquad\qquad \sigma_{\check{y}}^2 \approx \varepsilon_\mathrm{a}^2 + \varepsilon_\mathrm{r}^2 \check{x}^2 \tag{A11}$$

$$\hat{y} \approx \hat{x}, \qquad\qquad\qquad\qquad\qquad\qquad \sigma_{\hat{y}}^2 \approx \varepsilon_\mathrm{a}^2 + \varepsilon_\mathrm{r}^2 \hat{x}^2 \tag{A12}$$

$$\bar{y} \approx \bar{x}, \qquad\qquad\qquad\qquad\qquad\qquad \sigma_{\bar{y}}^2 \approx \frac{1}{n}\big(\varepsilon_\mathrm{a}^2 + \varepsilon_\mathrm{r}^2(\bar{x}^2 + s_x^2)\big), \tag{A13}$$

$$5 \quad s_y \approx \sqrt{\max\big\{s_x^2 - (\varepsilon_\mathrm{a}^2 + \varepsilon_\mathrm{r}^2 \bar{x}^2), 0\big\}}, \qquad\qquad \sigma_{s_y}^2 \geq \frac{1}{n}\big(\varepsilon_\mathrm{a}^2 + \varepsilon_\mathrm{r}^2(\bar{x}^2 + 3s_x^2)\big). \tag{A14}$$

## A2.4 Combined uncertainty

To arrive at a total uncertainty, we combine them using the combination rule for independent uncertainties from classical error propagation (Taylor, 1997):

$$\varepsilon_{\check{x}} = \sqrt{\tau_{\check{y}}^2 + \sigma_{\check{y}}^2}, \qquad\qquad \varepsilon_{\hat{x}} = \sqrt{\tau_{\hat{y}}^2 + \sigma_{\hat{y}}^2}, \qquad\qquad \varepsilon_{\bar{x}} = \sqrt{\tau_{\bar{y}}^2 + \sigma_{\bar{y}}^2}, \qquad\qquad \varepsilon_{s_x} = \sqrt{\tau_{s_y}^2 + \sigma_{s_y}^2}. \tag{A15}$$

Here we use $x$ instead of $y$ in the left-hand side subscripts because outside of this appendix there is no need to refer to the underlying model we use.

*Author contributions.* Erik Quaeghebeur performed the brunt of the work and wrote the paper. Michiel Zaaijer provided essential feedback on almost all aspects of the work in regular discussions, and so made sure errors were weeded out or avoided from the start. He also did revision work on the paper.

*Competing interests.* The authors declare that they have no competing interests.

*Acknowledgements.* This research is part of the Dutch EUROS program, which is supported by NWO domain Applied and Engineering Sciences and partly funded by the Dutch Ministry of Economic Affairs.

We thank Marc de Hoop (Mierij Meteo Systems) and Sicco Kamminga (Nortek) for providing useful information about various OWEZ instruments and Peter Eecen (ECN) for providing additional information about the measurement setup. In the context of our investigation of
the MMIJ dataset, extensive communication with people of ECN took place; we thank Hans Verhoef, Arno van der Werff, and Ingmar Alting for their time and effort, and Johan Peeringa and Clym Stock-Williams for facilitating this communication. Regarding FINO1, we thank Olaf Outzen (BSH: Bundesamt für Seeschifffahrt und Hydrographie), Markus Kreklau (BSH), and Richard Fruehmann (UL International GmbH - DEWI) for furnishing instrumentation information and for their replies to inquiries.

We thank two reviewers of an earlier version of this paper for their useful feedback, suggestions, and pointers. These have improved its
focus and coverage. We also thank the Wind Energy Science-reviewers Rémi Gandoin, Nikola Vasiljevic, and Hans Verhoef for their many valuable comments that have led to clear improvements, such as making available separate metadata files, introducing the project owner as a stakeholder, and considering the FAIR data principles.

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
