# Peer review of "How to improve the state-of-the-art in metocean measurement datasets"

_Wind Energy Science, 2019_

## Referee Comment (RC1) · Rémi Gandoin (Referee) · 4 Sep 2019

Review of article WES-2019-42 / Rémi Gandoin rga@c2wind.com

Dear authors, thank you for writing this interesting article. I have been kindly asked to review it and to provide my comments. I have done so directly into the pdf document, see the attached supplement (it includes typos and other minor comments). You will find below some general comments, as well as for each section of the paper a detailed description of the comments that are, in my opinion, the most important and that need be addressed.

**General comments:**

This article fits well within the scope of the WES journal, as it provides both a detailed technical insight into a key activity (measurements), provides some rigorous analytical approach to quantifying uncertainties, and concludes by making recommendations on how to improve measurement campaigns. The paper is long, but very interesting. I can relate to the work that was carried out, it followed some rigorous- and well documented steps, and provide a valid set of recommendations which could form the basis for data specifications scope of works on future campaigns. A number of aspects could be improved though, so as to best reach out to the readers of WES:

- ➢ A clearer definition of "data", and some more detailed information on the value of the data of each sensor given the different contexts (offshore wind projects, science, etc); see Comment 1.1,
- ➢ The inclusion of the project owner as a key stakeholder in the description of these measurement projects, and thereby a better definition of the word "data provider"; see Comment 1.2,
- ➢ Making references to the industry standards for the quantification of uncertainties, as well as to the scientific literature about sensor uncertainty (this need be done in particular for cup anemometers, in my opinion); see Comment 2.1,
- ➢ Revising the wording and the argumentation so as to maximise the chances for the set of recommendations to be implemented in future campaigns; See Comment 4.1.

**Specific comments:**

Section 1:

I have two main comments in this Section. The first one deals with the definition and use of metocean data, the second deals with the general setup of such measurement campaigns.

*Comment 1.1:* The term metocean data should be defined in this Section, as the word "data" could also be understood as model data, while the article deals with measurements. The text describing the measurement data could also include a longer list of analysis carried out using the datasets, and rank them by order of importance for research and commercial purposes. This, in my opinion, would provide a better frame for the argumentation in the paper. For instance for wind resource and design work, 10-minute wind speed- and direction measurements, turbulence intensity and ca. 30-minute spectral wave measurements, and currents, are by far some of the most important ones, and require extreme caution. On the other hand, precipitation or humidity sensors, and to some extent temperature sensors, are typically given much less attention. For research work, purposes may vary and thereby the paper could argue in favour or setting all measurement signals on equal foot, but this needs be made clear and be argumented. In effect, one could argue that priority should be given

to the sensors measuring the quantities at play in the phenomena that drive the marine boundary layer (i.e. momentum- and heat mean- and turbulent fluxes, sea surface roughness), and at the right temporal resolution (for turbulence spectra, 10-minute statistics are not enough for instance). Overall, this argumentation and hierarchisation of the measurements is missing, and could be provided by referring to specific sections of the international power performance- and design standards (respectively IEC61400-12 and IEC61300-3), and to the scientific literature (for instance [BEEKEN1]) where the original goals of these campaigns are described.

*Comment 1.2:* You write: "*In this section we also mention options for addressing issues described where it can be done compactly and where we believe it adds value for dataset producers*." I will come back to this at a later stage of the review, but it is important to single out the role of the project owner (i.e. the entity who ordered the campaign), which has a particular role in the project, that is: writing the technical scope of works, which then become part of the contract(s) between the project owner and the compan(ies) carrying out the campaign (mast designers, instrumentation providers, etc). To make it simple: contractors (it can be the measurement department from a university, or a private company), should legally do what has been agreed in the SoW, and their performance is measured accordingly. In the article, the term "data producer" is ambiguous, as it can refer both the entity making the data available, or the contractor in charge of the work, or both. This should be clarified, because a clear understanding of the split of responsibilities and of the interfaces between the actors, would help the reader argue for, and implement all the good ideas which you present in the article.

Section 2.

*Comment 2.1:* Section 2 starts with a detailed description of the data, then its processing and filtering, and finishes with Table 3 and a discussion on uncertainties after having presented both the inputs and methods used for obtaining these results. This is a very good structure. Just, I would like to point out two things:

First, the results would benefit from the framing of the argumentation mentioned in my comment 1.1 (i.e. what are the most important sensors and statistics, and how a possible error in one of the statistics would impact projects and research works). Once some relevant text added in Section 1, you could refer to it and add weight to the argumentation at the end of the Section (on the *TI* values for instance).

Second, the derivation of the uncertainty is done without referring to the industry standards, and also lacks sensor-specific (in particular for cups) references to the existing literature. I understand the article treats all sensors on equal foot and makes the same assumption (sum of two gaussian noise components, one driven by the absolute error and one relative error, both errors coming from the specification sheets); this approach is interesting but unfortunately a comparison with the existing industry standards is missing to make it relevant for the audience of this paper (scientists and practitioners in Wind Energy). If I take the example of the cup anemometer, which, as mentioned above, is by far the most important type of sensor on this mast for both research- and industry work:

> Its behaviour and uncertainties (including 10-minute standard deviation) are well described in a number of scientific papers (see [KRISTENSEN1] and [PINDADO1] for instance),

> ➢ The IEC61400-12 standards contain specific sections dealing with the uncertainty cup anemometer measurements, relying on the work carried out in test facilities (MEASNET) and in research labs (see [ACCUWIND] for instance),
>
> ➢ Specification sheets are rarely trusted 😊

By including these references to the literature, you will also be able to fine-tune the inputs to your uncertainty model. And, above all, you would help the reader by providing a comparison of the assumptions, methods and results obtained in the paper with the ones obtained using the industry standard methods and results, in particular for cup anemometers (which has been, and still is actually, a reference measurement sensor for many analyses).

As to the methods and results presented in the Appendix, I followed (with difficulty, towards the end) the maths but eventually also carried out a few tests using random wind time series of 10-minute durations. I checked that the results I obtained where of the same order of magnitude as in Table 3[1], and also checked that the modelled measured time series (true signal + noise) kept the same spectral properties (it could have been another kind of uncertainty to touch upon). You could provide such short examples in the form of a Python code, for the user to play with.

Section 4.

*Comment 4.1:* For the sake of the efficiency of your argumentation, as I mentioned in my comment in Section 1, I think this Section could be reformulated as:

a) The opportunities and benefits for all parties (project owner, contactor and analysis) of doing what you suggest,
b) The likely, underlying causes for the deficiencies you have observed (spoiler: a lot has to do with the scope of work),
c) The barriers to change and means to resolves them.

As mentioned in the "General comments" section, the tone of some phrases and expressions may neither be appropriate nor efficient for your purpose. Many have gone through the steps you describe, including myself. With time, doing so repeatedly project after project (these are wide-spread issues in geosciences and technologies in general) has moved us closer to the community of site investigations-, tests- and measurement professionals, their clients, and eventually to the standards requirements, assumptions and methods from our own community of wind & site conditions professionals. That is to say: solving the issues you bring up is a shared responsibility.

That being said, and once acknowledged that despite the great talent, efforts and energy deployed for making these offshore campaigns a success[2], "*Monitoring* [remains] *science's Cinderella*" [NISBET1]. That is: no grand plan is likely to help. The industry has its standards, and cannot really take-in more normative requirements in my opinion. There are already some informative guidelines about data sharing and good practices, you could also mention them, for instance see FAIR DATA [VASILJEVIC1]. Differences in quality and validity of the data should be made on a project-specific basis, within a limited, reasonable and cost-effective scope of work, as part of a contract between
* * *
[1] About this Table: please provide some simple numerical examples, so the reader understands clearly how to derive these uncertainty numbers using the absolute and relative error stated in Table1, but also the assumed sample mean and std used of the examples (at least for the cup).

[2] See videos of the FINO masts on https://www.youtube.com/user/FinoForschung/videos (for instance the nice "FINO2 Im Bann der Jahreszeiten" or the impressive "FINO3 - Sturm bei Einsatz mit Sattechniker"). In a completely different style, see these videos depicting the Fuhai met mast in Taiwan: https://www.youtube.com/watch?v=mr2q63GnRms, https://www.youtube.com/watch?v=3_sgP48zsnU.

two entities to which your work can contribute. I give an example from RVO about a recent floating LiDAR campaign in the pdf.

| [BEEKEN1] | FIVE YEARS OF OPERATION OF THE FIRST OFFSHORE WIND RESEARCH PLATFORM IN THE GERMAN. BIGHT – FINO1. Link: https://www.dewi.de/dewi/fileadmin/pdf/publications/Publikations/5_Beeken.pdf |
|---|---|
| [KRISTENSEN1] | The Perennial Cup Anemometer. Link: http://www.windsensor.com/application/files/9514/2694/4640/The_Perennial_Cup_Anemometer.pdf |
| [PINDADO1] | The Cup Anemometer, a Fundamental Meteorological Instrument for the Wind Energy Industry. Research at the IDR/UPM Institute. https://www.ncbi.nlm.nih.gov/pubmed/25397921 |
| [ACCUWIND] | ACCUWIND - Methods for classification of cup anemometers. https://orbit.dtu.dk/files/7703258/ris_r_1555.pdf. |
| [NISBET1] | Cinderella science. https://www.nature.com/articles/450789a |
| [VASILJEVIC1] | Exploring the FAIR data principles across disciplines: Case DTU Wind Energy. https://zenodo.org/record/1493874 |

**Rating**

From https://www.wind-energy-science.net/peer_review/review_criteria.html.

Scientific significance: Good (2). This is thanks to the derivation of uncertainties in Section 2, which the authors claim is original (I could not verify this within the review time). There is some value for the reader to understand the proposed approach, in my opinion.

Scientific quality: Good (2). The work is rigorous and reproduceable thanks to the code made available.

Presentation quality: Poor (4). I recommend to change the tone of some phrases and expressions (see my comments in the pdf), and also recommend to shorten the text where possible (the article is a little long, this and may discourage the readers having little time available to deep-dive into articles).

Additional questions:

1. *Does the paper address relevant scientific questions within the scope of WES?*
   Yes.
2. *Does the paper present novel concepts, ideas, tools, or data?*
   Yes.
3. *Is the paper of broad international interest?*
   Yes.
4. *Are clear objectives and/or hypotheses put forward?*
   Yes.
5. *Are the scientific methods valid and clear outlined to be reproduced?*
   Yes.
6. *Are analyses and assumptions valid?*

Most likely (numerical examples for uncertainty calculations, for ex. short python codes, would help the reader.

7. *Are the presented results sufficient to support the interpretations and associated discussion?*
   Yes.

8. *Is the discussion relevant and backed up?*
   Not entirely, because the paper is missing references and comparison to the existing industry standards on how to report and calculate uncertainties.

9. *Are accurate conclusions reached based on the presented results and discussion?*
   See no.8.

10. *Do the authors give proper credit to related and relevant work and clearly indicate their own original contribution?*
    Yes.

11. *Does the title clearly reflect the contents of the paper and is it informative?*
    See my comment in the pdf. Title may need be changed.

12. *Does the abstract provide a concise and complete summary, including quantitative results?*
    Yes.

13. *Is the overall presentation well structured?*
    Not entirely, see criteria "Presentation quality" above.

14. *Is the paper written concisely and to the point?*
    Not entirely, see criteria "Presentation quality" above.

15. *Is the language fluent, precise, and grammatically correct?*
    *Yes.*

16. *Are the figures and tables useful and all necessary?*
    *Yes.*

17. *Are mathematical formulae, symbols, abbreviations, and units correctly defined and used according to the author guidelines?*
    *Yes (a few typos though, see pdf).*

18. *Should any parts of the paper (text, formulae, figures, tables) be clarified, reduced, combined, or eliminated?*
    *No.*

19. *Are the number and quality of references appropriate?*
    *See no. 8.*

20. *Is the amount and quality of supplementary material appropriate and of added value?*
    *See no.6*

[revised manuscript text omitted]

$$\operatorname{Var}(s_X^2) = \operatorname{Var}(s_E^2) + 4\operatorname{Var}(\overline{yE} - \bar{y}\bar{E}) + 4\operatorname{Cov}(s_E^2, \overline{yE} - \bar{y}\bar{E}).$$

The last term of this expression is zero because all terms of its expansion contain odd powers of independent standard normal
20   random variables. We do not perform the tedious calculation of the first term, as it essentially expresses the uncertainty of the
measurement noise, which has been left unmodeled. Therefore we *ignore* this term, which means we consider a lower bound:

$$\frac{1}{4}\operatorname{Var}(s_X^2) \geq \operatorname{Var}(\overline{yE} - \bar{y}\bar{E})$$

$$= \operatorname{Var}(\overline{yE}) + \bar{y}^2\operatorname{Var}(\bar{E}) - 2\bar{y}\operatorname{
[revised manuscript text omitted]

---

## Referee Comment (RC2) · Nikola Vasiljevic (Referee) · 9 Oct 2019

**General comments**
This manuscript represent an exhaustive review of three metocean datasets. The review is well presented and provides important recommendations for both the data creators and data users. The manuscript does not fall under the category of traditional research paper. The manuscript can be categorized as a mix of data paper (newly emerging type of scientific communication) and as a classical review paper. I am accepting the publication of the paper if the authors take into account the following feedbacks and include them in their manuscript:

It is important that the authors communicate their finding by taking into account
the FAIR data principles. Accordingly, I would like them to try to score the FAIRness of three datasets before and after the implementation of their work. Use the following simple sheet to do this:
https://www.cessda.eu/content/download/3845/35038/file/20170707_How_FAIR_are_your_data_Jones.pdf

What would be also helpful in this work is to review the FAIRness analysis of DTU Wind Energy:
https://zenodo.org/record/1493874

It seems that authors are not aware of EERA-JP wind endorsed wind energy taxonomies, metadata and restricted vocabularies:
https://github.com/wind-energy/taxonomies-and-vocabularies
I request that these are introduced as a part of NetCDF metadata and also reviewed in their paper. Also, they can see / review the implementation of the taxonomies in DTU Data (data publishing platform of DTU):
https://data.dtu.dk/DTU_Wind_Energy

I am not sure about the authors background, but considering the tone authors use that somewhat 'prosecutes' data creators of these three datasets points out that the authors themselves are not experimenters, ie. people who are designing field campaigns, setting up measurement equipment in field, collecting and providing data. Otherwise the manuscript would contain discussion of potential reasons why the three datasets are in the condition as they are. My point here is that the authors need to make effort and understand and discuss the reasons why the original data creators did not spend additional time improving the datasets. Also, they need to stress out what would be a benefit general benefit for data producers to spend additional time/resources to improve their datasets? Why should they do it? Is there a reward for them? Why their managers should allocate resources to improve data? What their for

them? Otherwise the manuscript they are trying to publish will not have desired effect on the community.

This leads me to the comment related to the section of the paper 'Code and data availability'. If the original data creator allow you to published the improved datasets you must include them as authors of those new and improved datasets.

**Minor comments**
The following is the list of minor comments:
Pg 1 Ln 24 remove brackets
Pg 2 Ln 5 - 6 : 'In brief: (i) Yes, there are shared issues, but, not unexpectedly, not all of them in all datasets. (ii) Dataset producers can address the issues with a few non-onerous additions to their creation practice. ' - this text belongs to conclusion or discussion of the paper.
Pg 3 Ln 2-1 Rephrase sentence since it is not very clear what you are trying to say.
Pg 3 Ln 3 Provide url to the website
Pg 5 Ln 7 Again, provide url to the website
Pg 5 Ln 30 Faulty data and quality flags are interrelated
Pg 6 Ln 5 It is not clear whether you converted each dataset to both formats or not
Pg 6 Ln 7 Be more concrete regarding the automatic data issues detection
Pg 7 Ln 1 what is the 'normal range' for the series ? values please!
Figure 3 Axes don't contain any information, it is hard to understand plots. Include some of the text from Pg 8 in the figure caption.
Pg 11 Ln 26 Be aware what term accuracy really means. I think you are talking about the uncertainty
Pg 17 Ln 29 - 31 Include reference to the wind energy taxonomies, metadata and restricted vocabularies.
Pg 19 Ln 27 How did you help to fix the bug ?

---

## Referee Comment (RC3) · Hans Verhoef (Referee) · 15 Oct 2019

Summary

The authors successfully completed the tedious job of preparing for analysis met ocean datasets that came in different formats. As a result the article contains clear recommendations to the data set providers; the most important are to include metadata and to use netCDF. May netCDF be the standard in the geophysical and the meteorological worlds, and be common in the realms of universities and institutes, the reviewer doubts that is generally accepted in the commercial wind energy sector. Under developers there still may be a preference for plain rather than binary numbers. For the user of the data the article is on one hand of less interest; the main recommendation is to not

trust what you got, as many things could be wrong. But is this not what a user already is doing? This recommendation therefore is a collection of reminders. From another point of view it is interesting for a user if a standard data format would be used for the data sets in the sense that it would save time and therefore effort.

The article is technical rather than scientific in nature as it deals with things made by men. More importantly, the article is descriptive in nature and lacks a research question. Since it focusses on data preparation, It does not contain any analysis whatsoever, and that's a pity because what one likes to know is whether the met ocean climate differs between the three sites. (There are many studies on the met ocean climate in single spots (MMIJ, OWEZ, or Fino1) but not many which compare the climates in one or more station.) I am aware that this is not the scope for the authors but something which could be addressed.

Recommendation: Accept, after resolving minor issues: • There is a lot of statistics in the appendices A2.2 and A2.3. A reference (text book, own work, . . .) must be given. • The first three equations in line 28 on page 26 have the same right-hand sides. • The colors in fig 2 must be explained in the caption or the legend of that figure. (The explanation is in the text now.) • Think again about the use of uncommon words (non-onerous, relegate, gleaned, tuple). • For wind direction the authors mention North. It is not clear whether they mean compass North or not? • Figure 3 Missing labels • Figure 4 Missing labels • Figure 5 Missing labels • Figure 6 Missing labels • In the text it's mentioned that data can be downloaded. The urlsThe are not clear where to retrieve the data and the site http://www.WindOpZee.net should be mentioned since that is the location where MMIJ data can be retrieved. • The provided link in the article for Kouwenhoven is not valid anymore. It should behttps://www.noordzeewind.nl/nl_nl/kennis/meteogegevens/_jcr_content/par/iconlist/iconlistsection/link_stream/1554383838 manual-data-files-meteo-mast-noordzeewind.pdf I understand that it's difficult to have working links if people change the address ð§ŸŁ

In the article it's mentioned ECN (Energy research Centre of the Netherlands). Since

April 1, 2018 we are called ECN part of TNO and later this afternoon we will get another name. Can imagine that the authors can't keep track of all the changes in naming but just for information.

Again compliments to the authors to try to standardize the data formats!

———————————————

---

## Author Comment (AC1) · 29 Nov 2019

This response to reviews follows the general sectioning structure of the paper, but first the general comments are treated. The format is as follows:

- The reviewer comment is shown in an upright font.

  *Our response follows in blue and slanted font.*

  *The planned changes are written in red and slanted smaller font.*

If possible and appropriate, we will group comments of different reviewers to avoid repetition and improve brevity.

[Figure]

**General comments**

- Add project owner next to 'producers and users' as a stakeholder; they define the scope of work, where requirements can be specified and have an effect on the actual measurement campaign. Currently the term 'dataset producer' is used ambiguously. Relevant locations in the text: P2L20, P20L18, P22L4–5.

  *It is a good suggestion to include 'project owner', as this allows us to improve the global argument and indeed would make responsibilities clearer. Also, we should indeed be more precise about what we mean with 'dataset producer' and not unnecessarily mix in 'dataset provider', as this does not add a useful distinction for the paper's goals.*

  *We will rewrite the second paragraph of the introduction to explicitly introduce the 'project owner', 'dataset producer' (including its role as provider), and 'dataset user'. We will include 'project owner' as a recommendation recipient (cf. P2L20). We will change all occurrences of 'dataset provider' to 'dataset producer', to consistently use the terminology introduced for the different parties as delineated in the introduction. We will add recommendations for the 'project owner' as well, making it explicit that they are the ones that can actually enforce things (cf. P20L18). In the conclusions, we will make it clear that the project owner is the one that gives the dataset producer duties that would be beneficial to the dataset users.*

- There would be value added by discussing more uses for the datasets, including a deduced ranking of importance of specific quantities (e.g., wind speed being more important than humidity). At the very least a clear justification should be given if no such ranking is provided. The ranking could be initially given in the Introduction and worked out and used to guide the argumentation in Sect. 2.1.

  *We discuss the context and generality of our work in the fifth paragraph of the introduction. We state that it is relevant outside wind energy as well, but did not explicitly discuss whether we discriminate between the measured quantities.*

*We should make it explicit that our choice is to not focus on any specific subset of quantities. We feel that not doing so would make the paper more complex by adding another consideration and deviate from the actual approach we took during the research. It would force us to discuss matters that we have not sufficiently investigated and which would need far more work than is reasonable in the context of the revision of this paper.*

*We will add a sentence to that fifth paragraph to make our approach explicit. (We will, however, add the technical report of Beeken et al. mentioned by the reviewer at the end of Appendix A1.2.)*

- The tone of the paper is sometimes inconsiderate towards data creators. Examples: P7L13 'more intelligent handling', P20L23–24 '..., which ... the data', P21L15 'especially with ECN' may be read to imply the opposite for other parties, P21L16 'do the effort'.

*While a lack of consideration was not intended, we agree that the formulations come across as such and that this is not appropriate.*

*P7L13: 'more intelligent handling' will be replaced by 'more elaborate handling'. P20L23–24: We will removed 'which is most likely already available in your data management systems'. We will remove P21L15 'especially with ECN' (we express our gratitude to ECN in the acknowledgments). P21L16: 'do the effort' will essentially replaced by 'invest in'.*

- A reviewer feels that the paper lacks a research question.

*We can understand that judgment by the reviewer. The paper is not a classical research paper and that is reflected in the formulation of its goals. These goals are given in the Introduction (P2L4–6) as a pair of questions and brief answers that; we feel these are appropriate as a formulation of the research objectives.*

*No changes will be made.*

- Take into account FAIR data principles. Try to score the FAIRness of the datasets before and after implementation of the recommendations of the paper.

*Given the current interest in FAIRness, this may indeed be of interest to many readers. We can perform a FAIRness analysis, although more qualitative rather than quantitative in nature. (So without really scoring each dataset.)*

*We will add a whole new section that looks at the current status, then moves to what role the recommendations of this paper play in changing that status, to finally evaluate the role of the non-user stakeholders. We will refer to that Appendix at the end of the second introductory paragraph of Sect. 2.1.*

- Review the FAIRness analysis of DTU Wind Energy.

*This is an interesting overview of FAIRness activities at DTU. However, we feel that our paper is not the appropriate place to provide a review of this material.*

*No changes will be made.*

- Add EERA-JP wind energy metadata to the created binary datasets. Review the EERA-JP wind energy taxonomies, metadata, and vocabularies in the paper.

*We looked at the datasets on https:// data.dtu.dk/ DTU_Wind_Energy to find the proper way to add EERA-JP metadata to netCDF files (in a single attribute? in multiple attributes?), but found no example. We feel that our paper is not the appropriate place to provide a review of this material, just as we do not review, e.g., the CF Conventions.*

*We decided that we will add the metadata in a ‘EERA_JPWind’ attribute, with one descriptor per line:*

```
Activities:Measurements:Field Experiment
External Conditions:Location:Offshore:Offshore
External Conditions:Water Depth Category:Shallow Water
Data Categories:Meteorological
```
- A reviewer finds the paper to be a bit long and that it should be shortened where possible.

  *We agree that a shorter text is more accessible. But of course there is a trade-off between conciseness and amount of content. Our choice of content and what was placed in an appendix was deliberate. We do not see easy opportunities for shortening the length in a meaningful way. Therefore, without concrete pointers and arguments, we are not inclined to work on reducing the paper's length.*

  *No changes will be made.*

- Both 'off-shore' and 'offshore' are used; be consistent. (One reviewer suggests 'offshore'.)

  *We agree a single spelling should be used; 'offshore' seems preferred also by dictionaries.*

  *We will change all occurrences of 'off-shore' to 'offshore'.*

- Make sure the meaning of italic text is clear when used.

  *It is correct that we use italic text in different meanings: for emphasis, for foreign language names, and when introducing some concept/terminology. Of these* the journal's style guidelines *only allows the first two. We should of course follow the style guide. We assume that if we do, our usage may be considered sufficiently clear.*

  *We will follow the style guide and will remove italics for concept introduction, but sometimes add 'called' in front of the term introduced or quotes around it.*

- Avoid uncommon words such as 'non-onerous', 'relegate', 'gleaned', 'tuple'.

  *Judging what and what words would interfere with readers' reading is difficult. We will not make that judgment, but will change all but one of the words mentioned.*

*(We prefer 'tuple' over alternatives such as 'set' or 'vector' for precisely expressing what we want.)*

*We will change 'non-onerous' to 'non-burdensome', 'relegate to' to 'put in', and 'gleaned' to 'learned'.*

- ECN is mentioned in various places in the paper; this organization is in an unfinished state of name change.

  *Thank you for reminding us; it is useful for the readers to mention this.*

  *We will add a footnote mentioning that ECN is now part of TNO and that its name will change.*

**Frontmatter**

- The title should make it clear the paper treats measurement datasets. It is unclear who 'your' in the title refers to.

  *Making the title clearer and more specific is a good idea. (At the expense of being less catchy, perhaps.)*

  *We will replace "How to improve your metocean datasets" by "How to improve the state-of-the-art in metocean measurement datasets".*

**1   Introduction**

- Make it clear that 'data' can refer to both measurement as model data and that this paper discusses measurement data.
*It is true that talking about 'measurement data' makes things clearer. We do not think it necessary to mention and discuss 'model data' if it is clear we the paper is about measurement data.*

*P1L1: We will replace 'metocean datasets of 10-minute statistics' by 'datasets of 10-minute metocean measurement statistics'. P1L14: We will add a 'measurements' keyword. P1L16: We will replace 'data' by 'measurement data'. P20L27: We will replace 'metocean statistics datasets' by 'metocean measurement statistics datasets'. P21L20,P22L18: We will replace 'metocean datasets' by 'metocean measurement datasets'.*

- P1L17–18: Tower & substructure design and installation planning also need wave data. P1L19: Instruments are also placed on fixed offshore platforms.

  *We agree.*

  *We will integrate the suggested additions into the corresponding sentences.*

- P1L24: Integrate parenthetical in preceding sentence or remove parentheses. (Similar cases pointed out: P6L18–19, P8L26–27, P13L14–15.)

  *The sentences on P1L24, P6L18–19, P7L8–9, P8L26–27, P13L14–15, P13L17, P15L17–18, P16T3caption, P16L1–2, P19L10–12, and P27L16–17 are made parenthetical to de-emphasize them. This is appropriate according to the style advice we found online. Removing parentheses would remove this intentional de-emphasis. Shifting the parentheses into the preceding sentence would add a whole sentence as a parenthetical within that preceding sentence, which is stylistically strange (though appropriate for parenthetical material that is not a full sentence). We prefer to keep our current, intentional stylistic choice.*

  *No change will be made.*

- P2L5–6: One reviewer feels this sentence belongs in conclusions. Another explicitly mentions the surrounding P1L23–P2L9 is well-formulated.

*The content of the sentence is also present in the conclusions. We think that this 'preview' is useful for readers and feel supported concerning this by one reviewer.*

*No change will be made.*

- P2L14,18: Replace 'instructive' with 'instructional'.

  *Indeed, 'instructional' expresses intent, whereas 'instructive' expresses an effect.*

  *We will replace P2L14 "To achieve the informative and instructive goals of this paper, we [. . .]" with "We [. . .]" (so we will drop the first part of this sentence). We will replace P2L18 'instructive' with 'instructional'.*

- P2L17–18: Rephrase sentence (missing comma?).

  *Indeed, the sentence is a bit confusing as it is.*

  *We will add comma after 'described'.*

**2  The datasets and their analysis**

- P2L23: Is 'qua' a typo?

  *No, but it is uncommon and may be incorrect as used.*

  *We will replace it by 'in terms of'.*

**2.1  A first look at the datasets**

- Download URLs should be provided for the datasets analyzed.

  *Download URLs were included in the references, but this was not clear from the citations, as no year was included for these references.*
*We will add the (URL-visiting) year to to make download references stand out when citing.*

- A comparison of the metocean climates at the datasets' sites would have added value.

  *Such a comparison would indeed have value, but falls outside of the scope of this paper.*

  *No changes will be made.*

- P3L1–2: Sentence unclear.

  *It is not clear to us in what way the sentence is unclear.*

  *No change will be made.*

- Use 'FINO1' as used by BSH instead of 'FINO 1'.

  *It is correct that BSH uses 'FINO1'.*

  *We will use 'FINO1' throughout, including in the transformation scripts, except in Figure 1, for which the effort to change this would be disproportionate (for us).*

- Details about instruments (make, type) may provide added value.

  *This may indeed be useful to some readers. However, we feel that providing this information in the existing tables in the paper would make them too cluttered. Adding extra tables in an appendix is an option. However, this information is already available in the metadata we include in the transformed datasets and so in the transformation scripts. That is not the ideal location to reference, but if we move that out to separate metadata files, not only this, but also other metadata we have chosen not to include in the paper is made available conveniently.*

  *We will separate out the metadata from the scripts into separate human-readable and machine-readable YAML files included in the script bundle. We will mention this in the*
*'Code and data availability section' and at the end of the introductory paragraphs of Section 2.1.*

- It may provide added value to add the logger to the information provided about the measurement setups.

  *This may indeed be useful to some readers. The information available varies greatly between the three datasets. We feel that going into this in the paper would introduce too much detail that is not of interest to many readers. It is easy and possible to provide some information in metadata files in the script bundle.*

  *We will add files with some information about the loggers (make, type, number, reference) to the script bundle. We will mention this at the end of the introductory paragraphs of Section 2.1.*

- Make explicit what the provenance is of the uncertainty and range values given in the instrument & quantity tables.

  *That can indeed be of interest to some readers.*

  *We will add this information (for all datasets) to the metadata YAML files as comments.*

- P4T1: Specify orientation in degN instead of 'NE'. Kouwenhoven (2007) states a sampling frequency of 39 Hz for the ultrasonic anemometer instead of the listed 4 Hz. Perhaps make it clear that the thermometer and hygrometer are integrated into one instrument. Perhaps 'thermometer' is more correctly called 'temperature sensor'?

  *We agree that exact angles should be given.*
  *The sampling rate of the ultrasonic anemometer is 39 Hz, but this raw data is not sent to the logger; the output rate is 1 or 4 Hz (as can be seen in the spec sheet in Kouwenhoven's report) and I have seen the raw data files (a colleague had obtained access) and there were 2400 samples per 10 minutes, so 4 Hz.*
  *The thermometer and hygrometer are indeed two sensors integrated into one*

*package.*

*We feel that 'thermometer' is the correct term to use (in this case it is a resistance thermometer, but such detail is left for the metadata files); 'temperature sensor' may be interpreted to mean just the sensor part of the instrument and exclude the part that converts the quantity sensed to a numerical value.*

*We will give exact angles in the footnote 'ao' describing the orientation.*

*We will add a footnote 'i' indicating that the thermometer and hygrometer are two sensors integrated into one package to this table and also the others, where the same remark applies.*

*We will keep 'thermometer'.*

- Information from the original data file headers may provide value in the data file descriptions.

*This information is available in the instrument overview tables and in the metadata in the transformation scripts.*

*We will make no changes because of this comment. However, our separation of the metadata into separate YAML files will make all the information not in the tables more accessible, sufficiently so, we feel.*

- Use 'specification sheets' instead of 'spec sheets'. Use 'met. mast' instead of 'met-mast'.

*We have looked to actual usage (on-line) and think 'specification sheet' and 'met mast' are the most common ways of writing (although 'spec sheet' is also common).*

*We will change 'spec sheets' to 'specification sheets' and 'met-mast' to 'met mast' throughout the paper.*

- P5L23: FINO1 generated data after 2016.

*Indeed. This may be of interest to the readers.*

*We will add a phrase "measurements are still ongoing" to the corresponding appendix (A.1.2).*

**2.2 Dataset issues**

- Discuss the possible reasons for data quality issues in the datasets. (This is also relevant for Sect. 4.)

  *We write from the perspective of the dataset user, as indicated in the third paragraph of the introduction. Of course we have some ideas about some of the possible causes for data quality issues, but do not have the insight necessary to usefully discuss this. This would be very interesting, but should probably be done from the perspective of the dataset producer, so by a dataset producer. What we do is, however, provide tools (including code) that can help the dataset producer identify, explain, and eliminate some classes of quality issues. It is also important to make it clear that we realize that it is inevitable for faulty data to be present in the raw measurements. Our aim is to improve the processing of that data into datasets such as the ones studied.*

  *We will modify the first paragraph of Sect. 2.2.1 to make the last point made above explicit: "It is normal that the measured signals (raw data) contain faulty data. [. . .] The dataset producers deal with such faulty data, e.g., by removing it, when creating the datasets of statistics series we study. Nevertheless, each of the three datasets presented above contained remaining faulty data."*

- P5L30: Faulty data and quality flags are interrelated.

  *It is true that quality flags can be used to indicate possibly faulty data. This is a good idea to include.*

*We will add a paragraph at the end of Sect. 2.2.4: "Of course other information next to missingness mechanisms can be included in the quality flag bit field, also for non-missing values, as is done for FINO1. For example, this can be used to indicate possibly faulty data (cf. Sect. 2.2.1) that has not been removed (made missing)."*

- P6L5: Clarify which datasets were converted to which formats.

  *That can indeed be of interest to the readers and it can be convenient that they do not need to look this up in the referenced script bundle.*

  *We will change '(HDF5 or netCDF4)' to '(HDF5 format for OWEZ and netCDF4 format for MMIJ and FINO1)'.*

- P6L7: Be more concrete regarding the automatic data issues detection.

  *We elaborate on this in the list below this sentence. For even more concrete information, the scripts themselves are available. We think this is enough, but perhaps the reviewer had something else in mind.*

  *No change will be made.*

- The colors in Fig. 2 must be explained in the caption or the legend of that figure.

  *This information used to be closer to the plots themselves, but we were requested to move this in-text to better adhere to the WES style. However, we understand that such information may be useful.*

  *We will add a summary of the in-text explanation: "(Mean in black; mean $\pm$ standard deviation in blue; minimum and maximum in red.)".*

- P7L1: What is the 'normal range' for the series?

  *Our language use was sloppy here. We meant 'instrument's range'.*

  *We will replace 'normal range' by 'instrument's range'.*

**WESD**

- P7L6–7: Perhaps give numerical examples to illustrate the standard deviation bound.

*Given that this is a purely mathematical, rough bound that serves as a sanity check, specific examples are not really of interest. We had a look at the empirical distribution of $2s_x/|\hat{x} - \check{x}|$ for a variable (MMIJ 'TrueWs' at 92 m), but also that did not show anything interesting, i.e., almost all samples lie far below the bound (mean is 0.35, standard deviation is 0.05). We really think a numerical example for this will not provide added value.*

*No changes will be made.*

- P8L1–5: 'Max' is strange for this categorical variable; is the same issue present for 'Value'?

*The MMIJ dataset has all four statistics calculated from the raw values even for categorical variables. In Section 2.2.3 "Statistic Selection" we already comment on this. Therefore the 'avg/Value' column really contains averaged values and is not really useful (but that was not the focus of this part of the paper). The 'Max' column should only contain real samples and therefore it should not contain non-existent codes, even if the concept of maximum is not really applicable (strange). So it is still a useful column in the context of this part of the paper, i.e., checks for faulty (categorical) data.*

*No change will be made.*

- Axis labels are missing in Figs. 3–6. These figures are hard to understand without extra information in the captions.

*The axis labels were omitted consciously to not overburden the plots. The information necessary for understanding the figures (including axis meanings) used to be closer to the plots themselves, but we were requested to move this in-text to better adhere to the WES style. However, we understand that such information may be useful.*

*We will not add any axis labels. We will add the parenthetical "(cf. pages x–y for an explanation)" to each caption to point the reader to the explanation. (There is too much explanation to put a summary in each caption.)*

- 'North' is mentioned without clarifying whether it is geomagnetic or geographic.

  *North is only mentioned in the context of the MMIJ dataset. The reason is that the boom designation is offset from typical direction angles. In the documentation it is not mentioned whether this is geographic or magnetic North (or even a grid North), but we assume it is not magnetic, because that would be atypical.*

  *We will preface '(geographic)' in front of the two occurrences of 'North'.*

- P11L2: Move footnote superscript before comma.

  *The footnote refers to all material delimited by the comma the footnotemark is attached to and the preceding comma, not just to the parenthetical or a specific word. Therefore, placing the footnotemark there is appropriate according to style guides.*

  *No changes will be made.*

- P11L7–12: Another, uncommon issue is drift of the logger clock.

  *Yes, we have heard colleagues discuss this in the context of SCADA data. We have no indication this was an issue for these datasets.*

  *No changes will be made.*

- P11L13–17: Descriptions and drawings do not always reflect actual placement; pictures or videos of the mounted instruments are useful in this regard.

  *We can agree with that.*

  *We will add '(Pictures or video footage would of course further increase confidence in the accuracy of the drawings.)' after the first sentence of this paragraph.*

- P11L19: Isn't the precipitation detector mentioned on page 56 of ECN-Wind-Memo-12-010?

  *No, that is the precipitation monitor, a different instrument, which is well-documented.*

  *No changes will be made.*

- P11L26: The term 'accuracy' is used instead of 'uncertainty'.

  *We consciously chose to use 'accuracy' here, trying to follow the usage described by the JCGM and mentioned in footnote 1. As we understand JCGM's definitions, 'accuracy' is the qualitative counterpart to the quantitative 'uncertainty' and so uncertainty provides accuracy information. We do not know whether the reviewer means to say that he disagrees with our interpretation or is following another definition. We assume the latter for now.*

  *We will add a footnotemark to 'Accuracy' that refers to footnote 1, to clarify our usage of the term.*

- P12L4: Refer to Sect. 2.2.5 to make it clear why the sampling frequency is important.

  *That would be helpful indeed.*

  *We will add a reference to Sect. 2.2.5.*

- P12L29: Make explicit relative to what in 'relatively little effort'.

  *Making that explicit would indeed be helpful.*

  *We will use 'little effort relative to the whole of the measurement campaign'.*

- Sensor and quantity-specific treatment is missing (cf. comment about ranking of quantities). For the discussion of uncertainty, references to and comparisons with existing work (including industry standards) are lacking.

*As stated in the discussion of the 'General comments', we choose not to do a quantity-specific treatment. We also choose not to do an instrument-specific treatment. However, it would be useful to explicitly inform the reader that our treatment is generic and about the existence of such specific treatments. The reviewer suggests focusing on cup anemometers. He also provides texts about cup anemometers that can be used in the comparison.*

– *Of these, Kristensen's paper discusses and quantifies biases in the 10-minute averaged wind speed and suggests an approach to remove (much of) that bias using wind direction measurements; if we understand the results correctly, these biases are supplementary to the uncertainties we derive.*

– *Pindado et al.'s review presents dynamical models and empirical data, but as far as I can see, no explicit expressions for uncertainy.*

– *The standard IEC61400-12 discusses how per-wind speed bin absolute uncertainties should be calculated (Appendix F.8); these can form the basis for the absolute and relative uncertainties used in our procedure. It also discusses the uncertainty of wind speed (Appendix E.5.3) as a combination of component uncertainties, not all of which may be of interest to be included in the statistics datasets. I find the discussion in the standard to not be very clear about the impact the time interval the wind speed is averaged over; this makes positioning its procedure relative to ours difficult.*

*The first and last texts provide sufficient material to create a paragraph to inform the readers.*

We will add an extra paragraph at the end of Sect. 2.2.5: "Before closing this Section, it is important to stress that the expressions for propagated uncertainties and biases above are generic. Namely, their derivation does not depend on the specific quantity considered or instrument used. Detailed knowledge of the measuring instrument's properties may allow for better uncertainty estimates or additional uncertainty and bias terms. For
*example, for cup anemometers, it is known that there is a positive bias of 0.5%–8% in the mean wind speed, but that this bias can be greatly reduced using wind direction variance estimates (Kristensen, 1999). Also, the IEC 61400-12-1 standard prescribes how the wind speed uncertainty should be calculated for calibrated cup anemometers (IEC, 2017, App. F), which may lead to high-quality estimates for $\varepsilon_\mathrm{a}$ and $\varepsilon_\mathrm{r}$."*

- P14L31: $\varepsilon_{\bar{x}}^2$ instead of $\varepsilon_{\bar{x}^2}$.

  *It should indeed be $\varepsilon_{\bar{x}}^2$.*

  *We will correct this.*

- P15L17–18: A numerical example would add value for understanding the magnitude of the bias's effect. P16T3: More generally, numerical examples can clarify how the derivation of uncertainties is done. Perhaps such examples can or can also be provided as Python code.

  *Making the bias's effect more concrete would indeed add value. The derivation of the uncertainties is done as per the equations in this section, of course, and the code for generating the table should be made available to make it clear that effectively just that is done. It is not clear to us whether the reviewer would prefer more steps to be put in the paper's text (we do not think this would have sufficient added value).*

  *The code for generating the table's values will be included in the script bundle. To allow for easier interpretation, we will add an extra column to the table, for the relative value of the bias-corrected standard deviation. We will furthermore make our comment about the impact of the bias on turbulence intensity more concrete and add a comment about bias and uncertainty for ambient temperature.*

- Choose more formal or precise alternative word for 'bunch' (P10L1) and 'quite a lot' (P12L27). 'Timestamp' instead of 'Time stamp'.

*We disagreed amongst ourselves about 'bunch', so we will follow the reviewer's preference to replace it. Given its context, we think 'quite a lot' is fine here, i.e., the discussion above makes it explicit what we mean. Nevertheless, we think 'a good amount' would be a better formulation (not that it is more formal). Both 'Timestamp' and 'Time stamp' are in use, but we have no objections to your preference.*

*We will replace 'a bunch' by 'several' and 'a cluster', respectively. We will change 'quite a lot' to 'a good amount'. We will change 'Time stamp' to 'Timestamp'.*

**3 Dataset formatting**

3.1 A comparison of data file formats

- P17L7: Clarify what is called useless and why.

  *We meant to say that using a text editor for analysis is useless. This comment in the text is not essential and apparently not clear.*

  *We will remove 'and useless for analysis'.*

- P17L27: Rephrase to avoid quotes around 'knows'.

  *We agree that this formulation is not that good.*

  *We will change 'knows' to 'has access to'.*

- P17L29–31: Refer to EERA-JP wind energy taxonomies etc. here.

  *If we include terms from the taxonomy in our datasets, we should indeed also cite reference material.*

  *We will cite*

– *https:// github.com/ wind-energy/ taxonomies-and-vocabularies*
– *https:// doi.org/ 10.5281/ zenodo.1199488*

**3.2 Practical experiences with binary formats**

- P19L27: How did you help fix the buggy Python netCDF4 code?

  *We filed a bug report and actively assisted in getting it fixed. Actually, we did the same with another issue. However, I now think this paper is not the place to try and get credit for that.*

  *We will remove the remark about helping to fix the buggy code.*

**4 Recommendations**

- Give reasons for dataset creators to follow the recommendations. Sketch opportunities, barriers to change, and means to resolve them.

  *The main reason why dataset producers should follow the recommendations is because it would improve the usefulness for users of the datasets they deliver. This is already clear in the paper. But of course, even if we think this would cost relatively little effort, this costs time and therefore money. By introducing the project owner as a stakeholder (see discussion earlier), we can make it clear how following the recommendations can be fit into the agreed-on duties of the dataset producer. Also, the value of improved datasets to project owners as input to future measurement can then be mentioned. We do not think a wider discussion of opportunities and barriers to change falls within the scope of our paper. We have the perspective of the dataset user and gaining the necessary insight for such a discussion would for us be a project unto itself; we certainly do not wish to speculate on this.*

*We will add the following sentences to the last paragraph of the Conclusions: "This effort can be seen by the project owner as necessary for getting the most value out of the raw data collected. Such a well-documented dataset with uncertainty and quality information included creates the possibility for consciously making possibly different choices (trade-offs) when setting up future measurement campaigns.". No further changes will be made apart for those related to other comments involving the stakeholders (see "General comments") and their shared responsibility (see below).*

- A reviewer states that solving the issues discussed in the paper is a shared responsibility and that more normative requirements are not realistic.

    *We agree that this is a shared responsibility. The earlier suggested introduction of the project owner makes it possible to sketch the responsibilities of the stakeholders in the paper. We do not argue explicitly for more normative requirements, but would recommend project owners make certain concrete requirements for dataset producers. Our assessment is that the benefits of these outweigh their costs.*

    *In the recommendations, we will add some parentheticals specifically aimed at making the shared responsibility and each stakeholder's role clearer.*

- A reviewer contrasts the current acceptance of binary formats in academia with a preference for text-based formats in the commercial sector.

    *The last recommendation for users is relevant in this context: our experience shows that binary formats are much more efficient to work with and we have become convinced that this would be the case for almost any party, be it commercial or academic. We do think this is actually the most relevant recommendation in the list and needs to be made more prominent and forceful.*

    *We will move that last recommendation to the front and reformulate it to: "Invest in learning to work with format like HDF5 or netCDF4, as this will allow working more efficiently with datasets (cf. Sect. 3).".*

- A reviewer indicates that the recommendations to users are more reminders and that the main benefit for them would be the generalized use of a standardized time and effort-saving format.

  *We think the argument that binary formats can reduce time and effort spent by users is sufficiently made in the paper. The recommendations for users are indeed not as strong as those for producers. However, even the obvious recommendation about not trusting the data blindly must, we feel, be kept: in our own project there were mathematician/computer science researchers that use such datasets in a purely instrumental fashion, without an inclination to perform checks first.*

  *The change that will be made due to the point above this one already makes the recommendations a bit stronger. We will make a minor further improvement by also switching the order of the other two recommendations and by making it clear that we realize not trusting the data blindly would be obvious for many readers.*

- Keep the recommendations impersonal; avoid 'you'.

  *We have no clear preference here, so we will follow yours.*

  *We will reformulate the recommendations to remove 'you(r)'.*

- P20L26: Why the parentheses around 'also'?

  *To express that providing the dataset binary format can be done next to CSV file (or some such), but that we do not think providing the latter is necessary. But actually, this shouldn't be our concern and focus. Getting binary format files is, so we should just leave out the 'also'.*

  *We will drop '(also)'.*

- P21L4: Rounding to the expressed uncertainty would lead to a loss of information in case the uncertainty is revised downward.

*We think the metadata and data should be consistent. So if the information (metadata) used to determine the uncertainty of values is revised, then a revised dataset should be published, based on the reprocessed raw or intermediate data. If conservative estimates (lower bounds) of the uncertainties are used (e.g., as proposed in this paper), the revised datasets should in general not include uncertainty reduction. Because the difference between possible precision and actual precision is large in general, binary-rounding also leads to substantial space savings (after compression; non-significant digits are essentially random and do not compress well). Given all these reasons (and some others that would lead too far), we stick to our current recommendation of binary-rouding values.*

*We will add a recommendation "Use clear version identification in dataset files, to avoid confusion when updated or extended datasets are released.".*

- P21L6: Provide original sample standard deviation next to the bias-corrected version.

*In principle, we think this should not be done, for the same reasons as mentioned in the reply just above. However, we understand this feels like a more invasive change than rounding to uncertainty, even if there is a real error that is corrected by this procedure. Because of that, we mentioned the alternative option of not correcting but just including the bias values.*

*No changes will be made.*

**A Appendices**

- P25T2: Wasn't there a statistic labeled 'variance' in the FINO1 datasets?

*No, not in the version we downloaded. But BSH may have changed the files they make available; I think that the current version may even be different from the one*

*I downloaded and analyzed.*

*No changes will be made.*

- A reference is needed in support of the statistics-heavy material.

  *We understand that this part is not as accessible, but it is a bit hard for us to judge which statements require referencing.*

  *We will add three more specific citations to the standard text by Cramér, to support statements that may not be as well-known as we assumed. (The first author has a background in probability theory...)*

- The first three equations in line 28 on page 26 have the same right-hand sides.

  *Indeed, and that is correct.*

  *No changes will be applied.*

**Backmatter**

- In the list of references there are stale URLs and missing version and techreport numbers.

  *The reviewers are correct about the stale URLs and missing numbers.*

  *The changes to be made to the reference entries are therefore:*

  - *fix stale URLs,*
  - *remove unnecessary stale URLs,*
  - *update all software entries to the currently used version,*
  - *move all version numbers to title field,*
  - *make sure BibTeX entry types are chosen such as to expose the number field once compiled.*

---

## Editor Comment (EC1) · Andrea Hahmann (Editor) · 10 Dec 2019

Dear Authors,

Thanks for taking the time to answer the reviewer comments. However, the way the responses are organized is not appropriate. It is nearly impossible to check every reviewer's comment against your response. Could you please split the responses between the various reviewers? They can still be in the same document but organized by the reviewer. If the answer is the same to two or more comments, just write "as in response to RC1 Comment 2", for example.

Best regards, Andrea Hahmann

---

## Author Response (AR1)

**Response to reviews**

Erik Quaeghebeur and Michiel B. Zaaijer

November 30, 2019

This response to reviews and list of changes made follows the general sectioning structure of the paper, but first the general comments are treated. The format is as follows:

- The reviewer comment is shown in an upright roman font.

  *Our response follows in an italic font in blue.*

  *The resulting changes are written in a smaller, slanted roman font in red.*

If possible and appropriate, we will group comments of different reviewers to avoid repetition and improve brevity.

We will include a version of the paper where the changes made are clearly marked using (red) strikethrough for removals and (blue) squiggly underlining for addition. This was generated using the `latexdiff` tool, which cannot detect all changes, so we have also indicated those changes using pdf annotations (yellow highlighting).

**General comments**

- Add project owner next to 'producers and users' as a stakeholder; they define the scope of work, where requirements can be specified and have an effect on the actual measurement campaign. Currently the term 'dataset producer' is used ambiguously. Relevant locations in the text: P2L20, P20L18, P22L4–5.

  *It is a good suggestion to include 'project owner', as this allows us to improve the global argument and indeed would make responsibilities clearer. Also, we should indeed be more precise about what we mean with 'dataset producer' and not unnecessarily mix in 'dataset provider', as this does not add a useful distinction for the paper's goals.*

  *We rewrote the second paragraph of the introduction to explicitly introduce the 'project owner', 'dataset producer' (including its role as provider), and 'dataset user'. Included 'project owner' as a recommendation recipient (cf. P2L20). Changed all occurrences of 'dataset provider' to 'dataset producer', to consistently use the terminology introduced for the different parties as delineated in the introduction. We have now added recommendations for the 'project owner' as well, making it explicit that they are the ones that can actually enforce things (cf. P20L18). In the conclusions, we now make it clear that the project owner is the one that gives the dataset producer duties that would be beneficial to the dataset users.*

- There would be value added by discussing more uses for the datasets, including a deduced ranking of importance of specific quantities (e.g., wind speed being more important than humidity). At the very least a clear justification should be given if no such ranking is provided. The ranking could be initially given in the Introduction and worked out and used to guide the argumentation in Sect. 2.1.

*We discuss the context and generality of our work in the fifth paragraph of the introduction. We state that it is relevant outside wind energy as well, but did not explicitly discuss whether we discriminate between the measured quantities. We should make it explicit that our choice is to not focus on any specific subset of quantities. We feel that not doing so would make the paper more complex by adding another consideration and deviate from the actual approach we took during the research. It would force us to discuss matters that we have not sufficiently investigated and which would need far more work than is reasonable in the context of the revision of this paper.*

*We have added a sentence to that fifth paragraph: "Therefore, we treat all measured quantities on equal footing and do not focus on wind and wave data.". No further changes have been made. (However, we have added the technical report of Beeken et al. mentioned by the reviewer at the end of Appendix A1.2.)*

- The tone of the paper is sometimes inconsiderate towards data creators. Examples: P7L13 'more intelligent handling', P20L23–24 '..., which ...the data', P21L15 'especially with ECN' may be read to imply the opposite for other parties, P21L16 'do the effort'.

  *While a lack of consideration was not intended, we agree that the formulations come across as such and that this is not appropriate.*

  *P7L13: 'more intelligent handling' replaced by 'more elaborate handling'. P20L23–24: Removed 'which is most likely already available in your data management systems'. Removed P21L15 'especially with ECN' (we express our gratitude to ECN in the acknowledgments). P21L16: 'do the effort' essentially replaced by 'invest in' (more changes have been made here due to other comments).*

- A reviewer feels that the paper lacks a research question.

  *We can understand that judgment by the reviewer. The paper is not a classical research paper and that is reflected in the formulation of its goals. These goals are given in the Introduction (P2L4–6) as a pair of questions and brief answers that; we feel these are appropriate as a formulation of the research objectives.*

  *No changes made.*

- Take into account FAIR data principles. Try to score the FAIRness of the datasets before and after implementation of the recommendations of the paper.

  *Given the current interest in FAIRness, this may indeed be of interest to many readers. We can perform a FAIRness analysis, although more qualitative rather than quantitative in nature. (So without really scoring each dataset.)*

  *We have added a whole new section (Appendix A1.3 "FAIRness analysis") that looks at the current status, then moves to what role the recommendations of this paper play in changing that status, to finally evaluate the role of the non-user stakeholders. We refer to that Appendix at the end of the second introductory paragraph of Sect. 2.1.*

- Review the FAIRness analysis of DTU Wind Energy.

  *This is an interesting overview of FAIRness activities at DTU. However, we feel that our paper is not the appropriate place to provide a review of this material.*

  *No changes made.*

- Add EERA-JP wind energy metadata to the created binary datasets. Review the EERA-JP wind energy taxonomies, metadata, and vocabularies in the paper.

*We looked at the datasets on `https://data.dtu.dk/DTU_Wind_Energy` to find the proper way to add EERA-JP metadata to netCDF files (in a single attribute? in multiple attributes?), but found no example. We feel that our paper is not the appropriate place to provide a review of this material, just as we do not review, e.g., the CF Conventions.*

*We decided to add the metadata in a '`EERA_JPWind`' attribute, with one descriptor per line:*

```
Activities:Measurements:Field Experiment
External Conditions:Location:Offshore:Offshore
External Conditions:Water Depth Category:Shallow Water
Data Categories:Meteorological
```

- A reviewer finds the paper to be a bit long and that it should be shortened where possible.

  *We agree that a shorter text is more accessible. But of course there is a trade-off between conciseness and amount of content. Our choice of content and what was placed in an appendix was deliberate. We do not see easy opportunities for shortening the length in a meaningful way. Therefore, without concrete pointers and arguments, we are not inclined to work on reducing the paper's length.*

  *No changes made.*

- Both 'off-shore' and 'offshore' are used; be consistent. (One reviewer suggests 'offshore'.)

  *We agree a single spelling should be used; 'offshore' seems preferred also by dictionaries.*

  *Changed all occurrences of 'off-shore' to 'offshore'.*

- Make sure the meaning of italic text is clear when used.

  *It is correct that we use italic text in different meanings: for emphasis, for foreign language names, and when introducing some concept/terminology. Of these the journal's style guidelines only allows the first two. We should of course follow the style guide. We assume that if we do, our usage may be considered sufficiently clear.*

  *We now follow the style guide and have removed italics for concept introduction, sometimes adding 'called' in front of the term introduced or quotes around it.*

- Avoid uncommon words such as 'non-onerous', 'relegate', 'gleaned', 'tuple'.

  *Judging what and what words would interfere with readers' reading is difficult. We will not make that judgment, but will change all but one of the words mentioned. (We prefer 'tuple' over alternatives such as 'set' or 'vector' for precisely expressing what we want.)*

  *Changed 'non-onerous' to 'non-burdensome', 'relegate to' to 'put in', and 'gleaned' to 'learned'.*

- ECN is mentioned in various places in the paper; this organization is in an unfinished state of name change.

  *Thank you for reminding us; it is useful for the readers to mention this.*

  *We have added a footnote mentioning that ECN is now part of TNO and that its name will change.*

**Frontmatter**

- The title should make it clear the paper treats measurement datasets. It is unclear who 'your' in the title refers to.

*Making the title clearer and more specific is a good idea. (At the expense of being less catchy, perhaps.)*

Replaced "How to improve your metocean datasets" by "How to improve the state-of-the-art in metocean measurement datasets".

**1 Introduction**

- Make it clear that 'data' can refer to both measurement as model data and that this paper discusses measurement data.

  *It is true that talking about 'measurement data' makes things clearer. We do not think it necessary to mention and discuss 'model data' if it is clear we the paper is about measurement data.*

  P1L1: Replaced 'metocean datasets of 10-minute statistics' by 'datasets of 10-minute metocean measurement statistics'. P1L14: Added a 'measurements' keyword. P1L16: Replaced 'data' by 'measurement data'. P20L27: Replaced 'metocean statistics datasets' by 'metocean measurement statistics datasets'. P21L20,P22L18: Replaced 'metocean datasets' by 'metocean measurement datasets'.

- P1L17–18: Tower & substructure design and installation planning also need wave data. P1L19: Instruments are also placed on fixed offshore platforms.

  *We agree.*

  We have integrated the suggested additions into the corresponding sentences.

- P1L24: Integrate parenthetical in preceding sentence or remove parentheses. (Similar cases pointed out: P6L18–19, P8L26–27, P13L14–15.)

  *The sentences on P1L24, P6L18–19, P7L8–9, P8L26–27, P13L14–15, P13L17, P15L17–18, P16T3caption, P16L1–2, P19L10–12, and P27L16–17 are made parenthetical to de-emphasize them. This is appropriate according to the style advice we found online. Removing parentheses would remove this intentional de-emphasis. Shifting the parentheses into the preceding sentence would add a whole sentence as a parenthetical within that preceding sentence, which is stylistically strange (though appropriate for parenthetical material that is not a full sentence). We prefer to keep our current, intentional stylistic choice.*

  *No change made.*

- P2L5–6: One reviewer feels this sentence belongs in conclusions. Another explicitly mentions the surrounding P1L23–P2L9 is well-formulated.

  *The content of the sentence is also present in the conclusions. We think that this 'preview' is useful for readers and feel supported concerning this by one reviewer.*

  *No change made.*

- P2L14,18: Replace 'instructive' with 'instructional'.

  *Indeed, 'instructional' expresses intent, whereas 'instructive' expresses an effect.*

  Replaced P2L14 "To achieve the informative and instructive goals of this paper, we [...]" with "We [...]" (so we dropped the first part of this sentence). Replaced P2L18 'instructive' with 'instructional'.

- P2L17–18: Rephrase sentence (missing comma?).

  *Indeed, the sentence is a bit confusing as it is.*

  Added comma after 'described'.

**2 The datasets and their analysis**

- P2L23: Is 'qua' a typo?

  *No, but it is uncommon and may be incorrect as used.*

  We have replaced it by 'in terms of'.

**2.1 A first look at the datasets**

- Download URLs should be provided for the datasets analyzed.

  *Download URLs were included in the references, but this was not clear from the citations, as no year was included for these references.*

  We added the (URL-visiting) year to to make download references stand out when citing.

- A comparison of the metocean climates at the datasets' sites would have added value.

  *Such a comparison would indeed have value, but falls outside of the scope of this paper.*

  No changes made.

- P3L1–2: Sentence unclear.

  *It is not clear to us in what way the sentence is unclear.*

  No change made.

- Use 'FINO1' as used by BSH instead of 'FINO 1'.

  *It is correct that BSH uses 'FINO1'.*

  We now use 'FINO1' throughout, including in the transformation scripts, except in Figure 1, for which the effort to change this would be disproportionate (for us).

- Details about instruments (make, type) may provide added value.

  *This may indeed be useful to some readers. However, we feel that providing this information in the existing tables in the paper would make them too cluttered. Adding extra tables in an appendix is an option. However, this information is already available in the metadata we include in the transformed datasets and so in the transformation scripts. That is not the ideal location to reference, but if we move that out to separate metadata files, not only this, but also other metadata we have chosen not to include in the paper is made available conveniently.*

  We have separated out the metadata from the scripts into separate human-readable and machine-readable YAML files included in the script bundle. (This was a lot of work, but we think the increased accessibility of this metadata is worth it.) We now mention this in the 'Code and data availability section' and at the end of the introductory paragraphs of Section 2.1.

- It may provide added value to add the logger to the information provided about the measurement setups.

  *This may indeed be useful to some readers. The information available varies greatly between the three datasets. We feel that going into this in the paper would introduce too much detail that is not of interest to many readers. It is easy and possible to provide some information in metadata files in the script bundle.*

  We have added files with some information about the loggers (make, type, number, reference) to the script bundle. We mention this at the end of the introductory paragraphs of Section 2.1.

- Make explicit what the provenance is of the uncertainty and range values given in the instrument & quantity tables.

  *That can indeed be of interest to some readers.*

  We have added this information (for all datasets) to the metadata YAML files as comments. Also a few table values in the paper have been removed/added/changed as a consequence of digging up all the necessary information.

- P4T1: Specify orientation in degN instead of 'NE'. Kouwenhoven (2007) states a sampling frequency of 39 Hz for the ultrasonic anemometer instead of the listed 4 Hz. Perhaps make it clear that the thermometer and hygrometer are integrated into one instrument. Perhaps 'thermometer' is more correctly called 'temperature sensor'?

  *We agree that exact angles should be given.*
  *The sampling rate of the ultrasonic anemometer is 39 Hz, but this raw data is not sent to the logger; the output rate is 1 or 4 Hz (as can be seen in the spec sheet in Kouwenhoven's report) and I have seen the raw data files (a colleague had obtained access) and there were 2400 samples per 10 minutes, so 4 Hz.*
  *The thermometer and hygrometer are indeed two sensors integrated into one package.*
  *We feel that 'thermometer' is the correct term to use (in this case it is a resistance thermometer, but such detail is left for the metadata files); 'temperature sensor' may be interpreted to mean just the sensor part of the instrument and exclude the part that converts the quantity sensed to a numerical value.*

  We have given exact angles in the footnote 'ao' describing the orientation.
  We have added a footnote 'i' indicating that the thermometer and hygrometer are two sensors integrated into one package to this table and also the others, where the same remark applies.
  We have kept 'thermometer'.

- Information from the original data file headers may provide value in the data file descriptions.

  *This information is available in the instrument overview tables and in the metadata in the transformation scripts.*

  We have made no changes because of this comment. However, our separation of the metadata into separate YAML files makes all the information not in the tables more accessible, sufficiently so, we feel.

- Use 'specification sheets' instead of 'spec sheets'. Use 'met. mast' instead of 'met-mast'.

  *We have looked to actual usage (on-line) and think 'specification sheet' and 'met mast' are the most common ways of writing (although 'spec sheet' is also common).*

  We have changed 'spec sheets' to 'specification sheets' and 'met-mast' to 'met mast' throughout the paper.

- P5L23: FINO1 generated data after 2016.

  *Indeed. This may be of interest to the readers.*

  We have added a phrase "measurements are still ongoing" to the corresponding appendix (A.1.2).

**2.2 Dataset issues**

- Discuss the possible reasons for data quality issues in the datasets. (This is also relevant for Sect. 4.)

  *We write from the perspective of the dataset user, as indicated in the third paragraph of the introduction. Of course we have some ideas about some of the possible causes for data quality issues, but do not have the insight necessary to usefully discuss this. This would be very interesting, but should probably be done from the perspective of the dataset producer, so by a dataset producer. What we do is, however, provide tools (including code) that can help the dataset producer identify, explain, and eliminate some classes of quality issues. It is also important to make it clear that we realize that it is inevitable for faulty data to be present in the raw measurements. Our aim is to improve the processing of that data into datasets such as the ones studied.*

  We have modified the first paragraph of Sect. 2.2.1 to make the last point made above explicit: "It is normal that the measured signals (raw data) contain faulty data. [. . . ] The dataset producers deal with such faulty data, e.g., by removing it, when creating the datasets of statistics series we study. Nevertheless, each of the three datasets presented above contained remaining faulty data."

- P5L30: Faulty data and quality flags are interrelated.

  *It is true that quality flags can be used to indicate possibly faulty data. This is a good idea to include.*

  We have added a paragraph at the end of Sect. 2.2.4: "Of course other information next to missingness mechanisms can be included in the quality flag bit field, also for non-missing values, as is done for FINO1. For example, this can be used to indicate possibly faulty data (cf. Sect. 2.2.1) that has not been removed (made missing)."

- P6L5: Clarify which datasets were converted to which formats.

  *That can indeed be of interest to the readers and it can be convenient that they do not need to look this up in the referenced script bundle.*

  We have changed '(HDF5 or netCDF4)' to '(HDF5 format for OWEZ and netCDF4 format for MMIJ and FINO1)'.

- P6L7: Be more concrete regarding the automatic data issues detection.

  *We elaborate on this in the list below this sentence. For even more concrete information, the scripts themselves are available. We think this is enough, but perhaps the reviewer had something else in mind.*

  No change made.

- The colors in Fig. 2 must be explained in the caption or the legend of that figure.

  *This information used to be closer to the plots themselves, but we were requested to move this in-text to better adhere to the WES style. However, we understand that such information may be useful.*

  We have now added a summary of the in-text explanation: "(Mean in black; mean ± standard deviation in blue; minimum and maximum in red.)".

- P7L1: What is the 'normal range' for the series?

  *Our language use was sloppy here. We meant 'instrument's range'.*

  *Replaced 'normal range' by 'instrument's range'.*

- P7L6–7: Perhaps give numerical examples to illustrate the standard deviation bound.

  *Given that this is a purely mathematical, rough bound that serves as a sanity check, specific examples are not really of interest. We had a look at the empirical distribution of $2s_x/|\hat{x} - \check{x}|$ for a variable (MMIJ 'TrueWs' at 92 m), but also that did not show anything interesting, i.e., almost all samples lie far below the bound (mean is 0.35, standard deviation is 0.05). We really think a numerical example for this will not provide added value.*

  *No changes made.*

- P8L1–5: 'Max' is strange for this categorical variable; is the same issue present for 'Value'?

  *The MMIJ dataset has all four statistics calculated from the raw values even for categorical variables. In Section 2.2.3 "Statistic Selection" we already comment on this. Therefore the 'avg/Value' column really contains averaged values and is not really useful (but that was not the focus of this part of the paper). The 'Max' column should only contain real samples and therefore it should not contain non-existent codes, even if the concept of maximum is not really applicable (strange). So it is still a useful column in the context of this part of the paper, i.e., checks for faulty (categorical) data.*

  *No change made.*

- Axis labels are missing in Figs. 3–6. These figures are hard to understand without extra information in the captions.

  *The axis labels were omitted consciously to not overburden the plots. The information necessary for understanding the figures (including axis meanings) used to be closer to the plots themselves, but we were requested to move this in-text to better adhere to the WES style. However, we understand that such information may be useful.*

  *We did not add any axis labels. We have added the parenthetical "(cf. pages x–y for an explanation)" to each caption to point the reader to the explanation. (There is too much explanation to put a summary in each caption.)*

- 'North' is mentioned without clarifying whether it is geomagnetic or geographic.

  *North is only mentioned in the context of the MMIJ dataset. The reason is that the boom designation is offset from typical direction angles. In the documentation it is not mentioned whether this is geographic or magnetic North (or even a grid North), but we assume it is not magnetic, because that would be atypical.*

  *We have prefaced '(geographic)' in front of the two occurrences of 'North'.*

- P11L2: Move footnote superscript before comma.

  *The footnote refers to all material delimited by the comma the footnotemark is attached to and the preceding comma, not just to the parenthetical or a specific word. Therefore, placing the footnotemark there is appropriate according to style guides.*

  *No changes made.*

- P11L7–12: Another, uncommon issue is drift of the logger clock.

*Yes, we have heard colleagues discuss this in the context of SCADA data. We have no indication this was an issue for these datasets.*

*No changes made.*

- P11L13–17: Descriptions and drawings do not always reflect actual placement; pictures or videos of the mounted instruments are useful in this regard.

  *We can agree with that.*

  *Added '(Pictures or video footage would of course further increase confidence in the accuracy of the drawings.)' after the first sentence of this paragraph.*

- P11L19: Isn't the precipitation detector mentioned on page 56 of ECN-Wind-Memo-12-010?

  *No, that is the precipitation monitor, a different instrument, which is well-documented.*

  *No changes made.*

- P11L26: The term 'accuracy' is used instead of 'uncertainty'.

  *We consciously chose to use 'accuracy' here, trying to follow the usage described by the JCGM and mentioned in footnote 1. As we understand JCGM's definitions, 'accuracy' is the qualitative counterpart to the quantitative 'uncertainty' and so uncertainty provides accuracy information. We do not know whether the reviewer means to say that he disagrees with our interpretation or is following another definition. We assume the latter for now.*

  *We have added a footnotemark to 'Accuracy' that refers to footnote 1, to clarify our usage of the term.*

- P12L4: Refer to Sect. 2.2.5 to make it clear why the sampling frequency is important.

  *That would be helpful indeed.*

  *Added a reference to Sect. 2.2.5.*

- P12L29: Make explicit relative to what in 'relatively little effort'.

  *Making that explicit would indeed be helpful.*

  *We now use 'little effort relative to the whole of the measurement campaign'.*

- Sensor and quantity-specific treatment is missing (cf. comment about ranking of quantities). For the discussion of uncertainty, references to and comparisons with existing work (including industry standards) are lacking.

  *As stated in the discussion of the 'General comments', we choose not to do a quantity-specific treatment. We also choose not to do an instrument-specific treatment. However, it would be useful to explicitly inform the reader that our treatment is generic and about the existence of such specific treatments. The reviewer suggests focusing on cup anemometers. He also provides texts about cup anemometers that can be used in the comparison.*

  - *Of these, Kristensen's paper discusses and quantifies biases in the 10-minute averaged wind speed and suggests an approach to remove (much of) that bias using wind direction measurements; if we understand the results correctly, these biases are supplementary to the uncertainties we derive.*

  - *Pindado et al.'s review presents dynamical models and empirical data, but as far as I can see, no explicit expressions for uncertainy.*

– *The standard IEC61400-12 discusses how per-wind speed bin absolute uncertainties should be calculated (Appendix F.8); these can form the basis for the absolute and relative uncertainties used in our procedure. It also discusses the uncertainty of wind speed (Appendix E.5.3) as a combination of component uncertainties, not all of which may be of interest to be included in the statistics datasets. I find the discussion in the standard to not be very clear about the impact the time interval the wind speed is averaged over; this makes positioning its procedure relative to ours difficult.*

*The first and last texts provide sufficient material to create a paragraph to inform the readers.*

We have added an extra paragraph at the end of Sect. 2.2.5: "Before closing this Section, it is important to stress that the expressions for propagated uncertainties and biases above are generic. Namely, their derivation does not depend on the specific quantity considered or instrument used. Detailed knowledge of the measuring instrument's properties may allow for better uncertainty estimates or additional uncertainty and bias terms. For example, for cup anemometers, it is known that there is a positive bias of 0.5%–8% in the mean wind speed, but that this bias can be greatly reduced using wind direction variance estimates (Kristensen, 1999). Also, the IEC 61400-12-1 standard prescribes how the wind speed uncertainty should be calculated for calibrated cup anemometers (IEC, 2017, App. F), which may lead to high-quality estimates for $\varepsilon_{\mathrm{a}}$ and $\varepsilon_{\mathrm{r}}$."

- P14L31: $\varepsilon_{\bar{x}}^2$ instead of $\varepsilon_{\bar{x}^2}$.

  *It should indeed be $\varepsilon_{\bar{x}}^2$.*

  Corrected.

- P15L17–18: A numerical example would add value for understanding the magnitude of the bias's effect. P16T3: More generally, numerical examples can clarify how the derivation of uncertainties is done. Perhaps such examples can or can also be provided as Python code.

  *Making the bias's effect more concrete would indeed add value. The derivation of the uncertainties is done as per the equations in this section, of course, and the code for generating the table should be made available to make it clear that effectively just that is done. It is not clear to us whether the reviewer would prefer more steps to be put in the paper's text (we do not think this would have sufficient added value).*

  The code for generating the table's values is now included in the script bundle. While doing so, we discovered a mistake ($\max\{\sqrt{\ldots}, 0\}$ instead of $\sqrt{\max\{\ldots, 0\}}$), which we corrected; this correction has shown the biases are even more pronounced. To allow for easier interpretation, we have added an extra column to the table, for the relative value of the bias-corrected standard deviation. We have furthermore made our comment about the impact of the bias on turbulence intensity more concrete and added a comment about bias and uncertainty for ambient temperature.

- Choose more formal or precise alternative word for 'bunch' (P10L1) and 'quite a lot' (P12L27). 'Timestamp' instead of 'Time stamp'.

  *We disagreed amongst ourselves about 'bunch', so we will follow the reviewer's preference to replace it. Given its context, we think 'quite a lot' is fine here, i.e., the discussion above makes it explicit what we mean. Nevertheless, we think 'a good amount' would be a better formulation (not that it is more formal). Both 'Timestamp' and 'Time stamp' are in use, but we have no objections to your preference.*

  Replaced 'a bunch' by 'several' and 'a cluster', respectively. Changed 'quite a lot' to 'a good amount'. Changed 'Time stamp' to 'Timestamp'.

**3  Dataset formatting**

**3.1  A comparison of data file formats**

- P17L7: Clarify what is called useless and why.

  *We meant to say that using a text editor for analysis is useless. This comment in the text is not essential and apparently not clear.*

  *We have removed 'and useless for analysis'.*

- P17L27: Rephrase to avoid quotes around 'knows'.

  *We agree that this formulation is not that good.*

  *Changed 'knows' to 'has access to'.*

- P17L29–31: Refer to EERA-JP wind energy taxonomies etc. here.

  *If we include terms from the taxonomy in our datasets, we should indeed also cite reference material.*

  *Cited*

  - *https://github.com/wind-energy/taxonomies-and-vocabularies*
  - *https://doi.org/10.5281/zenodo.1199488*

**3.2  Practical experiences with binary formats**

- P19L27: How did you help fix the buggy Python netCDF4 code?

  *We filed a bug report and actively assisted in getting it fixed. Actually, we did the same with another issue. However, I now think this paper is not the place to try and get credit for that.*

  *Removed the remark about helping to fix the buggy code.*

**4  Recommendations**

- Give reasons for dataset creators to follow the recommendations. Sketch opportunities, barriers to change, and means to resolve them.

  *The main reason why dataset producers should follow the recommendations is because it would improve the usefulness for users of the datasets they deliver. This is already clear in the paper. But of course, even if we think this would cost relatively little effort, this costs time and therefore money. By introducing the project owner as a stakeholder (see discussion earlier), we can make it clear how following the recommendations can be fit into the agreed-on duties of the dataset producer. Also, the value of improved datasets to project owners as input to future measurement can then be mentioned. We do not think a wider discussion of opportunities and barriers to change falls within the scope of our paper. We have the perspective of the dataset user and gaining the necessary insight for such a discussion would for us be a project unto itself; we certainly do not wish to speculate on this.*

  *We have added the following sentences to the last paragraph of the Conclusions: "This effort can be seen by the project owner as necessary for getting the most value out of the raw data collected. Such a well-documented dataset with uncertainty and quality information included creates the possibility for consciously making possibly different choices (trade-offs) when setting up future*

*measurement campaigns.".* No further changes made apart for those related to other comments involving the stakeholders (see "General comments") and their shared responsibility (see below).

- A reviewer states that solving the issues discussed in the paper is a shared responsibility and that more normative requirements are not realistic.

  *We agree that this is a shared responsibility. The earlier suggested introduction of the project owner makes it possible to sketch the responsibilities of the stakeholders in the paper. We do not argue explicitly for more normative requirements, but would recommend project owners make certain concrete requirements for dataset producers. Our assessment is that the benefits of these outweigh their costs.*

  In the recommendations, we have now added some parentheticals specifically aimed at making the shared responsibility and each stakeholder's role clearer.

- A reviewer contrasts the current acceptance of binary formats in academia with a preference for text-based formats in the commercial sector.

  *The last recommendation for users is relevant in this context: our experience shows that binary formats are much more efficient to work with and we have become convinced that this would be the case for almost any party, be it commercial or academic. We do think this is actually the most relevant recommendation in the list and needs to be made more prominent and forceful.*

  Moved that last recommendation to the front and reformulated it to: "Invest in learning to work with format like HDF5 or netCDF4, as this will allow working more efficiently with datasets (cf. Sect. 3).".

- A reviewer indicates that the recommendations to users are more reminders and that the main benefit for them would be the generalized use of a standardized time and effort-saving format.

  *We think the argument that binary formats can reduce time and effort spent by users is sufficiently made in the paper. The recommendations for users are indeed not as strong as those for producers. However, even the obvious recommendation about not trusting the data blindly must, we feel, be kept: in our own project there were mathematician/computer science researchers that use such datasets in a purely instrumental fashion, without an inclination to perform checks first.*

  The change made due to the point above this one already makes the recommendations a bit stronger. We have made a minor further improvement by also switching the order of the other two recommendations and by making it clear that we realize not trusting the data blindly would be obvious for many readers.

- Keep the recommendations impersonal; avoid 'you'.

  *We have no clear preference here, so we will follow yours.*

  Reformulated the recommendations to remove 'you(r)'.

- P20L26: Why the parentheses around 'also'?

  *To express that providing the dataset binary format can be done next to CSV file (or some such), but that we do not think providing the latter is necessary. But actually, this shouldn't be our concern and focus. Getting binary format files is, so we should just leave out the 'also'.*

  We have dropped '(also)'.

- P21L4: Rounding to the expressed uncertainty would lead to a loss of information in case the uncertainty is revised downward.

  *We think the metadata and data should be consistent. So if the information (metadata) used to determine the uncertainty of values is revised, then a revised dataset should be published, based on the reprocessed raw or intermediate data. If conservative estimates (lower bounds) of the uncertainties are used (e.g., as proposed in this paper), the revised datasets should in general not include uncertainty reduction. Because the difference between possible precision and actual precision is large in general, binary-rounding also leads to substantial space savings (after compression; non-significant digits are essentially random and do not compress well). Given all these reasons (and some others that would lead too far), we stick to our current recommendation of binary-rouding values.*

  *We added a recommendation "Use clear version identification in dataset files, to avoid confusion when updated or extended datasets are released.".*

- P21L6: Provide original sample standard deviation next to the bias-corrected version.

  *In principle, we think this should not be done, for the same reasons as mentioned in the reply just above. However, we understand this feels like a more invasive change than rounding to uncertainty, even if there is a real error that is corrected by this procedure. Because of that, we mentioned the alternative option of not correcting but just including the bias values.*

  *No changes made.*

**A   Appendices**

- P25T2: Wasn't there a statistic labeled 'variance' in the FINO1 datasets?

  *No, not in the version we downloaded. But BSH may have changed the files they make available; I think that the current version may even be different from the one I downloaded and analyzed.*

  *No changes made.*

- A reference is needed in support of the statistics-heavy material.

  *We understand that this part is not as accessible, but it is a bit hard for us to judge which statements require referencing.*

  *We have added three more specific citations to the standard text by Cramér, to support statements that may not be as well-known as we assumed. (The first author has a background in probability theory...)*

- The first three equations in line 28 on page 26 have the same right-hand sides.

  *Indeed, and that is correct.*

  *No changes applied.*

**Backmatter**

- In the list of references there are stale URLs and missing version and techreport numbers.

  *The reviewers are correct about the stale URLs and missing numbers.*

  *The changes to the reference entries are therefore:*

- fixed stale URLs (squiggly blue underline in annotated pdf),
- removed unnecessary stale URL (entry highlighted in yellow in annotated pdf),
- updated all software entries to the currently used version,
- moved all version numbers to title field,
- made sure BibTeX entry types are chosen such as to expose the number field once compiled.

[revised manuscript text omitted]
_{\mathrm{a}}^2 \mathbb{E}(Z_{\mathrm{a},k}^2) + 2\varepsilon_{\mathrm{a}}\varepsilon_{\mathrm{r}} y_k \mathbb{E}(Z_{\mathrm{a},k})\mathbb{E}(Z_{\mathrm{r},k}) + \varepsilon_{\mathrm{r}}^2 y_k^2 \mathbb{E}(Z_{\mathrm{r},k}^2)\big) = \varepsilon_{\mathrm{a}}^2 + \varepsilon_{\mathrm{r}}^2 \bar{y}^{(2)},$$

$$\mathbb{E}(\bar{E}^2) = \frac{1}{n^2}\sum_{k=1}^{n}\sum_{\ell=1}^{n}\Big(\varepsilon_{\mathrm{a}}^2 \mathbb{E}(Z_{\mathrm{a},k}Z_{\mathrm{a},\ell}) + \varepsilon_{\mathrm{a}}\varepsilon_{\mathrm{r}}\big(y_k \mathbb{E}(Z_{\mathrm{a},\ell})\mathbb{E}(Z_{\mathrm{r},k}) + y_\ell \mathbb{E}(Z_{\mathrm{a},k})\mathbb{E}(Z_{\mathrm{r},\ell})\big) + \varepsilon_{\mathrm{
[revised manuscript text omitted]

---

## Author Response (AR2)

**Response to 2nd round review**

Erik Quaeghebeur and Michiel B. Zaaijer

January 17, 2020

We would like to thank the reviewer for his serious effort to go over all the changes in our revision. We very much appreciate his collegial attitude that helps to improve our paper.

This response to this second-round review and list of changes made follows the general sectioning structure of the paper. The format is as follows:

- The reviewer comment is shown in an upright roman font.

  *Our response follows in an italic font in blue.*

  *The resulting changes are written in a smaller, slanted roman font in red.*

We will include a version of the paper where the changes made are clearly marked using (red) strikethrough for removals and (blue) squiggly underlining for addition. This was generated using the `latexdiff` tool, which cannot detect all changes, so we have also indicated those changes using pdf annotations (yellow highlighting).

**1 Introduction**

- P1L18: I suggest you write "wind turbine support structure", as it is the terminology in the Figure 1-1 of IEC61400-3-1. To the terms "tower" and "substructure", you would otherwise have to add "foundation".

  *Good suggestion.*

  *Replaced 'tower & substructure design' by 'wind turbine support structure design'.*

**2 The datasets and their analysis**

**2.1 A first look at the datasets**

- P3L3: Sentence unclear; maybe repeat "we" between "other two" and "relegate".

  *Good suggestion. (N.B.: In our revision, we had already replaced 'relegate' by 'put [. . . ]')*

  *Replaced 'for the other two put' by 'for the other two we put'.*

- Consider adding the vertical datum (MSL or LAT, typically) to the metadata files.

  *For MMIJ (LAT), this is already available in the definition of the height coordinates in the file `structure.yaml`. For FINO1 (LAT), this is currently defined in the transformation scripts, and so is present in the dataset file generated, but not in the separate metadata files. For OWEZ (MSL), this is not currently present anywhere in the metadata.*

  *For FINO1, we have added the vertical datum information to `metadata-instruments.yaml` as a comment next to the 'measurement_height' attribute. For OWEZ, we have added a 'height' attribute to the channel metadata in `metadata-channels.yaml`.*

**2.2 Dataset issues**

- P6L6: Replace 'normal' by 'not unusual'.

  *Good suggestion.*

  *Done.*

- P6L7–8: Regarding "by removing it, when creating the datasets of statistics series we study". This is often not the case, as the user typically wants "all" the data, and applies his/her own filter. Maybe rephrase this using conditional tense.

  *In the datasets of our study, faulty data (detected by the producers) was mostly removed (replaced by NaN or another missing data indicator). However, in FINO1 data, also flags were used to indicate datapoint quality. This approach is compatible with the wish to get 'all' the data and would therefore also best not go unmentioned. This part of the paper is descriptive and not prescriptive, so we feel that a conditional 'could' instead of 'is' would not be appropriate.*

  *Replaced 'removing' by 'flagging or removing'.*

- P7L14–17: Regarding the upper bound for standard deviations, over a 10 minute period, the time series is not stationary, that is: is often has a trend, because of some mesoscale wind pattern (wind ramp for instance) that has a time scale longer than 10 min. Therefore, in some instances, the standard deviation of the wind speed over 10 min can be very large, but in these situations do not represent short term variations above and below the mean value of a stationary process. Therefore, the criteria you have chosen may flag as "faulty" a measurement which is not wrong, but would for example need to be detrended. Actually, it is common practice to detrend time series, for detailed analysis of turbulence; maybe could you mention this? Suggested reference: `https://orbit.dtu.dk/en/publications/de-trending-of-turbulence-measurements-2`.

  *In this part of the text, we discuss inconsistency of a set of statistics. These are violations of the mathematical relations that exist between the definitions of the statistics. The first criterion/example we give is that the sample mean should be between the sample minimum and sample maximum. The second criterion, the one you refer to, is the upper bound for the sample standard deviation in terms of the sample minimum and maximum. This criterion holds for any correctly calculated sample standard deviation (so also the case you describe) and is meant to detect problems with the calculation. After rereading the surrounding paragraph, which ends with the explicit "Any such inconsistency is a serious issue, as it indicates a deficiency somewhere in the procedures for calculating statistics and their post-processing.", we feel that no change is necessary.*

  *No change made.*

- Please also mention in the article that loggers are an essential piece of the measurement chain, and that logger programs/documentations need be checked and documented thoroughly.

  *It may indeed be a good idea to make this explicit.*

  *On page 12, at the end of the topic 'Instruments & their settings', we have added the following sentences: "Furthermore, loggers are an essential piece of the measurement chain and therefore need to be documented as well. For MMIJ and FINO1 this is the case, but not for OWEZ."*

- P16L3: Add the corresponding mean wind speed and standard deviations to the reported 20% error in turbulence intensity (the `x_avg` and `x_std` in your code).

*The reported 20% is the (dataset) average of the turbulence intensity reduction/bias (as stated in the text). There are no single* `x_avg` *and* `x_std`/`x_std_unbiased` *that correspond to it. It is possible to calculate dataset averages of these quantities as well; they are 9.46, 0.672/0.614, respectively (in m/s). However, because of the nonlinear relationship with TI and the dependence between them, it would be a stretch to say that these dataset averages correspond to that average TI bias. Concretely, the 'TI bias' calculated from these values would be $(1 - 0.614/0.672) \approx 8.5\%$, quite a bit lower than 'about 20%' (i.e., the 19% from the first row's last column value). Because of this, it would cause confusion to give (the dataset averages of) the quantities you ask for without quite a bit of elaboration. We feel that such elaboration would lead to far.*

*No change made.*

**4 Recommendations**

- P22L15–16: Because most loggers produce text files as "raw" material (only rarely do they produce binary files), the end-user will, most likely, always want to get hold of this original material. [. . . ] Time teaches the practitioner to never trust anything but the raw, primary files.

  *The availability of the original, raw data files is mostly orthogonal to the usefulness of binary formats. Dataset producers could in principle provide literal conversions of text-based raw data to a binary format (which support text fields as well, if even a conversion to floating-point would have too much impact), preserving many of the binary format's advantages. Of course, if the user does not trust the producer to do this appropriately for their purposes, the original files are necessary. But in that case, the user would profit from doing their own conversion to a binary format. (We feel no change to the text is required, as we see the reviewer's comment as compatible with the recommendation referred to.)*

  *No change made.*

[revised manuscript text omitted]